# Impact of Southern Ocean surface conditions on deep ocean circulation at the LGM: a model analysis

Fanny Lhardy[1], Nathaëlle Bouttes[1], Didier M. Roche[1,2], Xavier Crosta[3], Claire Waelbroeck[4], and Didier Paillard[1]

[1]Laboratoire des Sciences du Climat et de l'Environnement, LSCE/IPSL, CEA-CNRS-UVSQ-Université Paris-Saclay, F-91198 Gif-sur-Yvette, France
[2]Vrije Universiteit Amsterdam, Faculty of Science, Cluster Earth and Climate, de Boelelaan 1085, 1081HV Amsterdam, The Netherlands
[3]Univ. Bordeaux, CNRS, EPOC, EPHE, UMR 5805, F-33600 Pessac, France
[4]Laboratoire d'Océanographie et du Climat: Expérimentation et Approches Numériques (LOCEAN), IPSL, Université Pierre et Marie Curie, Paris, France

**Correspondence:** Fanny Lhardy (fanny.lhardy@lsce.ipsl.fr)

**Abstract.** Changes in water mass distribution are considered to be a significant contributor to the atmospheric $CO_2$ concentration drop to around 186 ppm recorded during the Last Glacial Maximum (LGM). Yet simulating a glacial Atlantic Meridional Overturning Circulation (AMOC) in agreement with paleotracer data remains a challenge, with most models from previous Paleoclimate Modelling Intercomparison Project (PMIP) phases showing a tendency to simulate a strong and deep North Atlantic Deep Water (NADW) instead of the shoaling inferred from proxy records of water mass distribution. Conversely, the simulated Antarctic Bottom Water (AABW) is often reduced compared to its pre-industrial volume, and the Atlantic Ocean stratification is underestimated with respect to paleoproxy data. Inadequate representation of surface conditions, driving deep convection around Antarctica, may explain inaccurately simulated bottom water properties in the Southern Ocean. We investigate here the impact of a range of surface conditions in the Southern Ocean in the iLOVECLIM model, using nine simulations obtained with different LGM boundary conditions associated with the ice sheet reconstruction (e.g. changes of elevation, bathymetry, and land-sea mask), and/or modelling choices related to sea-ice export, formation of salty brines, and freshwater input. Based on model-data comparison of sea-surface temperatures and sea ice, we find that only simulations with a cold Southern Ocean and a quite extensive sea-ice cover show an improved agreement with proxy records of sea ice, despite systematic model biases in the seasonal and regional patterns. We then show that the only simulation which does not display a much deeper NADW is obtained by parameterizing the sinking of brines along Antarctica, a modelling choice reducing the open ocean convection in the Southern Ocean. These results highlight the importance of the representation of convection processes, which have a large impact on the water mass properties, while the choice of boundary conditions appears secondary for the model resolution and variables considered in this study.

# 1 Introduction

The Southern Ocean is a major climate player. Due to its specific geographical setting, it acts as a heat exchanger and buffer between the south polar regions and the subtropics, but also connects the other oceanic basins. Furthermore, it is one of the few oceanic regions where deep water formation takes place. Indeed, cold surface temperatures and brine rejection consecutive to sea-ice formation allow for a large and localized density increase of surface waters triggering deep convection. As a result, the dense southern-sourced Antarctic Bottom Water (AABW) fills the bottom of the world ocean. Density gradients between this water mass and others – such as its counterpart, the northern-sourced North Atlantic Deep Water (NADW) – determine the water mass distribution and the large-scale circulation. Rearrangement of water masses explains part of past changes in the carbon storage capacity of the oceans (Buchanan et al., 2016; Khatiwala et al., 2019; Yu et al., 2016), which stresses the importance of correctly simulating the processes affecting the deep ocean circulation.

Multimodel studies using outputs from previous Paleoclimate Modelling Intercomparison Project (PMIP) phases showed that models simulate different responses of the Atlantic Meridional Overturning Circulation (AMOC) to the same Last Glacial Maximum (LGM) experimental design. Only a minority of PMIP2 models produce a shoaling of the NADW (Otto-Bliesner et al., 2007; Weber et al., 2007), while most PMIP3 models produce an intensified and deepened NADW (Muglia and Schmittner, 2015), at odds with reconstructions from paleotracer data which display a shallower NADW along with a denser, more voluminous and possibly more sluggish AABW during the last glacial compared to pre-industrial (PI) and modern times (Curry and Oppo, 2005; Howe et al., 2016). Models rarely simulate bottom water temperatures and salinities close to the ones suggested by the few pore-fluid measurements in the deep glacial Atlantic (Adkins et al., 2002; Otto-Bliesner et al., 2007). Moreover, Heuzé et al. (2013) showed that, even in present-day conditions, models generally simulate inaccurate bottom water temperatures, salinities and densities. Even when they do simulate relatively accurate modern bottom water properties, they tend to form AABW via the wrong process (namely open ocean deep convection) whereas the largest proportion of AABW currently results from brine-dominated formation of dense shelf waters, overflowing in the deep ocean (Orsi et al., 1999; Williams et al., 2010). While some high resolution CMIP6 models now simulate dense shelf waters, Heuzé (2021) observed no obvious export of these waters, and open ocean deep convection remains a much too widespread and frequently occurring process.

As both the sea-surface temperature (SST) and salinity related to sea-ice formation in the Southern Ocean influence the surface density and therefore the AABW formation and properties, any surface conditions bias has the potential to impact the deep ocean circulation. Studies on the historical period have underlined important model biases in the Southern Ocean SSTs (Hyder et al., 2018) and sea ice (Downes et al., 2015), which could also affect paleoclimate simulations. And indeed, PMIP models struggle to reproduce the glacial sea-ice extent suggested by sea-ice proxy data, and especially its seasonality (Roche et al., 2012; Goosse et al., 2013; Marzocchi and Jansen, 2017). While Ferrari et al. (2014) have shown a dynamical link between the deep ocean circulation and Antarctic sea ice, Shin et al. (2003) have highlighted the major role played by Antarctic sea ice on the glacial AMOC by quantifying the haline density flux increase at the LGM in the CCSM model. Moreover, Marzocchi and Jansen (2017) have quantitatively attributed part of the observed discrepancies of the AMOC simulated by PMIP3 models to insufficient sea-ice formation and export. Therefore, targeting sea-ice biases in models may be necessary to improve the

simulated water mass distribution. It is also crucial to better understand and simulate the interplay between surface and deep conditions, especially as some processes – such as brine rejection (Bouttes et al., 2010) and downsloping currents (Campin and Goosse, 1999) – are inherently limited by the resolution of the models.

In this study, we use an intermediate complexity model under PMIP2 or PMIP4 experimental design and several bathymetries to generate a set of simulations computed with different boundary conditions. In addition to these simulations displaying contrasted surface conditions, three sensitivity tests of Southern Ocean conditions for sea-ice export, formation of brines, and freshwater input further document the role of sea ice on the deep ocean circulation. This variety of simulations allows us to investigate the respective effects of the many possible choices for boundary conditions and other experimental settings on the simulated surface conditions and associated deep water formation. We hereafter focus on Southern Ocean surface conditions and evaluate them using proxy data for both SSTs and sea ice. We rely on the principle of a simplified inverse methodology: we assess what improves the simulated temperatures and sea ice in the Southern Ocean, as evaluated against proxy data, and we analyse the associated impact on deep ocean circulation.

## 2   Methods

### 2.1   Model description

The iLOVECLIM model is a coupled Earth System Model of intermediate complexity (Claussen et al., 2002). Its relatively low computation time allows us to run multiple simulations, and to test the effect of different modelling choices and boundary conditions on surface conditions. Over time, iLOVECLIM has significantly diverged from its parent model LOVECLIM (Goosse et al., 2010), but is still composed in its core of an atmospheric component (ECBilt), a simple land vegetation model (VECODE) and an oceanic general circulation model (CLIO). With 20 irregular vertical levels and a horizontal resolution of $3° \times 3°$, CLIO is able to simulate the large-scale circulation, which is of interest to us in this study. It also includes a thermodynamic-dynamic sea-ice component described by Fichefet and Morales Maqueda (1997). This component simulates a visco-plastic rheology but no sea-ice thickness distribution, which is relatively classic compared to other PMIP models (see Table 1 of Goosse et al. (2013)) but far from the complexity of more recently developed sea-ice components (Rousset et al., 2015).

### 2.2   The PMIP boundary conditions and their implementation

The Paleoclimate Modelling Intercomparison Project (PMIP) provides standardized boundary conditions for paleoclimate simulations, enabling robust multimodel comparisons for periods of interest such as the LGM. The atmospheric gas concentrations and orbital parameters are prescribed to set values (e.g. a forcing parameter of 186 ppm for the glacial $CO_2$ concentration), based on data from Bereiter et al. (2015), Loulergue et al. (2008), Schilt et al. (2010), and Berger (1978). Since the ice sheet reconstructions are still associated with large uncertainties, Kageyama et al. (2017) describe the common experimental design for LGM experiments in the current phase 4 of the project but let modelling groups choose from three different ice sheet

reconstructions: GLAC-1D (Tarasov et al., 2012), ICE-6G-C (Peltier et al., 2015; Argus et al., 2014), or PMIP3 (Abe-Ouchi et al., 2015). To see the impact of such a choice, we have implemented in this study the boundary conditions (e.g. elevation, bathymetry, land-sea mask) associated with the first two options since these reconstructions are the most recent. We have also considered the results obtained with the previous LGM version of the model (PMIP2) described in Roche et al. (2007), which was generated with the boundary conditions associated with ICE-5G (Peltier, 2004), a previous reconstruction with notably higher elevation of the Northern Hemisphere ice sheets.

We have implemented the elevation associated with either GLAC-1D or ICE-6G-C topography at 21,000 years ago on the T21 grid of ECBilt in the Northern Hemisphere. The bathymetry of the CLIO grid has been modified according to the same topography and to a low sea level of -133.9 m (Lambeck et al., 2014). The bathymetry of the previous LGM version of iLOVECLIM was manually generated, while we now use a semi-automated method for our PMIP4 runs (see more detailed explanations in Appendix A). The land-sea mask is computed using the hypsometry discretized on the CLIO grid. A grid cell is defined as ocean if, at the subgrid level, the fraction of this cell below sea level exceeds a set threshold of 40%. The land-sea mask is manually defined in a few key regions (Gibraltar strait, Greenland–Iceland–Scotland threshold, etc.). We take particular care of this step, relying on knowledge of the sea level change and of the straits geography at the LGM, and also running a connectivity program (see Appendix B) computing sills to make informed choices.

## 2.3 Set of simulations

Thanks to the implementation of the PMIP boundary conditions and to the related development of the model, we have a set of five LGM simulations (Table 1) displaying contrasted climates (Fig. 1). Indeed, we ran two simulations under the PMIP4 experimental design ('P4-G' and 'P4-I', both also used in Kageyama et al. (accepted, 2021)) and three under the PMIP2 one ('New P2', 'Cold P2' and 'Warm P2'). We used different boundary conditions and/or modelling choices to obtain them. The boundary conditions (elevation, bathymetry, and land-sea mask) associated with the GLAC-1D, ICE-6G-C, or ICE-5G topography were implemented to obtain 'P4-G', 'P4-I' and 'New P2' respectively. The elevation associated with the ICE-5G topography was also implemented for the other two PMIP2 simulations ('Cold P2' and 'Warm P2'), but with the former manual bathymetry instead of the one generated using our new semi-automated method. Finally, we made different modelling choices with respect to the glacial temperature profiles used in the radiative code of ECBilt for these last two simulations ('Cold P2' and 'Warm P2'). Indeed, due to the coarse vertical resolution of ECBilt, the model uses GCM vertical linearizations which are region-dependent. We kept the default continental profile in the first case and used the Greenland profile for all ice-covered regions in the Northern Hemisphere in the second one, resulting in a large difference in the global mean temperature of these two simulations.

We added to this set three sensitivity tests. The boundary conditions associated with ICE-6C-G were arbitrarily chosen as standard in these tests, which is why the simulation 'P4-I' is considered as a LGM reference in the following sections. Sensitivity tests using the simulation 'P4-G' as reference (i.e. GLAC-1D boundary conditions) yield fairly similar results (not shown here). In 'P4-I wind', we multiplied – in the Southern Ocean only – the meridional wind tension on ice by a coefficient of 3 in order to boost the sea-ice export in the Southern Hemisphere and therefore explore the possible impact of the Antarctic

sea-ice dynamics. We ran 'P4-I brines' using the parameterization of the sinking of brines described by Bouttes et al. (2010). The objective of this parameterization is to account for the sinking of dense water rejected during sea-ice formation. Indeed, this process is often limited by the horizontal resolution of models, as the rejected salt tends to get diluted in the surface grid cells where sea ice is forming. This parameterization allows for a fraction of the salt content of the surface grid cell to be transferred to the deepest grid cell underneath the location of sea-ice formation. As a result, the salinity and density of the bottom cells increase while the salinity and density of the surface grid cells decrease, without congruent motion of water masses. The modification of the salinity depends on the rate of sea-ice formation, as well as the chosen fraction parameter. Here the fraction was chosen at 0.8 to allow for a large effect of this sensitivity test, but the gradual effect of this parameter choice on the streamfunction is shown in Fig. S5, as well as the impact of this parameterization on the PI streamfunction (and deep water mass properties, see 'PI brines' simulation in Fig. S6). This simple parameterization is relatively different than a downsloping current one as it is not confined to the continental slope and does not create mixing along the way of the sinking brines. While "this brine mechanism is idealized, it reflects the impact of intense Antarctic sea-ice formation during the LGM" (Bouttes et al., 2010) on the AABW density. In contrast to this transfer of salt, an addition of a freshwater flux (of 0.6 Sv) around Antarctica was done in the 'P4-I hosing' hosing experiment, as described by Roche et al. (2010).

The simulations are briefly described in Table 1. Each simulation has been run either 3000 or 5000 years to ensure a quasi-equilibrium state. The drift for any individual simulation is less that $2 \times 10^{-4}$ °C per century for the deep ocean temperature (global mean of all oceans below 2,000 meters depth). The last 100 years are analyzed. We use this set of simulations to a) compare the simulated sea-surface temperatures and sea-ice extent to their distribution in the Southern Ocean inferred from data and b) explore the impact of these surface conditions on deep ocean circulation.

## 2.4 Experimental data

The simulated surface conditions are first compared with the LGM sea-surface temperatures reconstructed by MARGO Project Members (2009). Thanks to the use of multiple proxies (diatoms, radiolaria, dinoflagellates, foraminifera, Mg/Ca, and alkenones), this dataset, combining 696 individual records, provides a synthesis of our knowledge of the LGM ocean surface temperature. However, it should be noted that most proxies are calibrated against summer SST (Esper and Gersonde, 2014; Cortese and Prebble, 2015) or annual SST (Sikes et al., 1997; Prahl et al., 2000). Only planktonic foraminifera allow for the estimation of winter SST (Howard and Prell, 1992) but their growth is hampered, and restricted to a couple of species, south of the Polar Front (Bé and Hutson, 1977). As such, there are only few winter SST estimates to compare with the simulated ones. As for the model-data comparison of the PI SSTs, we relied on the modern WOA data (World Ocean Atlas, 1998) since it is the one used by MARGO Project Members (2009).

Secondly, to evaluate the glacial Antarctic sea-ice distribution we compiled sea-ice proxy data from Gersonde et al. (2005), Allen et al. (2011), Ferry et al. (2015), Benz et al. (2016), Xiao et al. (2016), Nair et al. (2019) and Ghadi et al. (2020). In this compilation, LGM data include three types of proxies: a quantitative proxy of yearly sea-ice duration, a quantitative proxy of the winter (September) or summer (February) sea-ice concentration, and finally a qualitative proxy (based on the relative abundance of diatoms *Fragilariopsis curta* + *F. cylindrus* for winter sea-ice presence and *F. obliquecostata* for summer sea-

ice presence). To integrate these different types of measurements, an index is built based on the number of proxies agreeing on the sea-ice presence (ranging from 0 to 3 in winter and 0 to 2 in summer, with halved values when a proxy is not very conclusive). Presence of sea ice at a given location is accepted when the qualitative or quantitative value is above the error on the calibration step (Gersonde and Zielinski, 2000; Crosta et al., 2004; Esper and Gersonde, 2014). Taking into account all marine cores, we draw the likely delimitation of sea-ice presence in austral winter. Unfortunately, there are too few proxy data available to robustly constrain the location of the austral summer sea-ice edge. We thus extrapolated the modern relationship between summer sea-ice extent and SST, whereby summer sea ice lies south of the 0°C isotherm (Nicol et al., 2000) to the LGM. Caution is therefore needed when using the results as this summer contour is not well-constrained.

We then estimated the sea-ice extent inferred from this data compilation: we imported these contours on a $360 \times 360$ points grid (of $1° \times 0.5°$ in longitude and latitude), computed the surface area contained within (summing the weighted area of each grid cell on a perfect sphere) and subtracted an estimated surface of the Antarctic continent (i.e. land and grounded ice sheet areal extent) at the LGM. Results are discussed in Sect 3.3. We estimated a glacial Antarctica of $16.8 \times 10^6$ km$^2$ by computing the total area of the continent and of the continental shelves (up to -1000 m) on a high resolution ($16 \times 16$ km) modern topographic dataset (Fretwell et al., 2013). This value falls close to a GIS surface area estimate of $16.4 \times 10^6$ km$^2$ using Bentley et al. (2014) Antarctic maps at 20 ka on a Lambert projection. To put this value into perspective, the modern Antarctic continent has a surface area of $13.9 \times 10^6$ km$^2$ (Fretwell et al., 2013), due to a smaller areal extension of the Antarctic ice sheet and a higher sea level. For the indicative error in the sea-ice surface extent computed, we have chosen the values of 15% (in winter) and 30% (in summer) for two reasons. First of all, it is difficult to estimate the uncertainty linked to the extrapolation of the sea-ice edges using marine core data, and it makes sense for this uncertainty to be larger in summer than in winter due to the scarcity of data. Secondly, another uncertainty is arising from the subtracted surface area of Antarctica at the LGM, which affects the estimated sea-ice extent (but not its seasonality). Its continental limit is speculative in some regions (Bentley et al., 2014), while the discretisation of this limit as a land-sea mask on a coarse resolution grid may induce an additional error. More precisely, with the ICE-6G-C and the GLAC-1D topographic files (with their $1080 \times 2160$ and $360 \times 360$ points grid resolutions respectively), we find a 21 ka Antarctic surface of $15.0 \times 10^6$ km$^2$ and $17.1 \times 10^6$ km$^2$ respectively. An uncertainty of this order of magnitude (2 millions of square kilometers) represents 6% and 20% of the sea-ice extent estimated in winter and summer respectively. If we further discretise the contours of the winter and summer sea-ice edges and of the ICE-6G-C Antarctic continent on the $3° \times 3°$ CLIO grid, we underestimate the sea-ice extent by $3.4 \times 10^6$ km$^2$ (in winter) and $1.7 \times 10^6$ km$^2$ (in summer), that is to say by 10% and 16% respectively. Considering the order of magnitude of these alternative estimates, error bars of 15% and 30% seem reasonable. Still, these estimates are only indicative of the order of magnitude of the error.

Finally, to also evaluate the simulated PI sea-ice extent, we used sea-ice data on the period 1979–2010 from Parkinson and Cavalieri (2012), who computed a mean extent of $18.5 \times 10^6$ km$^2$ (in September) and $3.1 \times 10^6$ km$^2$ (in February) – though it should be noted that the sea-ice extent we simulated in our pre-industrial run is not fully comparable with these modern values because of climate change over the last century.

## 3  Results

### 3.1  Global mean surface air temperature anomaly

Six out of eight of our runs display a global mean surface air temperature anomaly (LGM mean SAT – PI mean SAT) in the range of -4 ± 0.8°C (Fig. 1) estimated by Annan and Hargreaves (2013), though three of them fall close to its upper limit. The average climate of 'Cold P2' is too cold and 'P4-I hosing' is too warm to agree with this range. With a LGM cooling of around -3.3°C, we also note that the PMIP4 boundary conditions (with lower ice sheets compared to PMIP2) lead to a significantly warmer climate than the PMIP2 boundary conditions (see 'P4-G' and 'P4-I' compared to 'New P2'). Compared to other PMIP4 models, iLOVECLIM simulates a quite warm glacial climate, in agreement with previous evaluations (Roche et al., 2007): Kageyama et al. (accepted, 2021) shows that half of the PMIP4 models simulate a LGM cooling in the -3.7°C to -4°C range, while three colder models simulate a larger global SAT anomaly (up to -6.8°C). We note that the LGM mean SAT anomaly was recently re-evaluated at -6.1 ± 0.4 °C (Tierney et al., 2020), due to lower SAT in the tropics than previously reconstructed. Both iLOVECLIM and most of the other PMIP4 models simulate relatively modest SAT anomalies which do not compare well with such a large LGM mean SAT anomaly. Nonetheless, this estimation was obtained thanks to a field reconstruction of LGM temperatures using data assimilation in the CESM model, an innovative method which is not freed from potential model biases, CESM being the coldest model out of the PMIP4 ensemble in Kageyama et al. (accepted, 2021).

### 3.2  Sea-surface temperatures

Figure 2 shows that our set of simulations yields a variety of sea-surface temperatures, with some significant regional differences. The pre-industrial SSTs are obviously warmer than the ones simulated by the reference LGM simulation 'P4-I', with a marked anomaly in the North Atlantic and in the Southern Ocean (Fig. 2a). Overall, the three PMIP2 simulations show colder SSTs than 'P4-I' (Fig. 2b,c,d). The differences between 'P4-G' and 'P4-I' are small (Fig. 2e), with the exception of the eastern Atlantic and western Indian sectors of the Southern Ocean, south of the African continent, where 'P4-G' displays warmer SSTs. This positive anomaly is related to a southward shift of the Antarctic Circumpolar Current. Larger differences exist between 'P4-I' and its sensitivity tests, especially in the North Atlantic and in the Southern Ocean. We note that the transfer of salt to the bottom of the ocean leads to a cooling of the Southern Ocean ('P4-I brines', Fig. 2f), while the opposite occurs with the addition of a freshwater flux around Antarctica ('P4-I hosing', Fig. 2h). Observed in ice-free regions (i.e. where the SSTs are not necessarily at the freezing point value), this cooling is probably a consequence of the enhanced stratification, since a well-mixed water column in upwelling regions would tend to dampen the effect of low winter surface temperatures on the SSTs. The third sensitivity test ('P4-I wind') only yields small differences with 'P4-I', except around Kerguelen Islands. A latitudinal gradient along the Atlantic is sometimes visible in the SST anomalies ('New P2', 'P4-I wind'), suggesting a change in the meridional heat transport, possibly due to the influence of the choice of boundary conditions and of the sensitivity tests on the AMOC (see Sect. 3.4).

We now explore which of these surface conditions agree best with the proxy data from MARGO Project Members (2009). To quantify the model-data agreement, we compute the root mean square errors (RMSEs) for each ocean basin, for both the

austral summer (JFM) and winter (JAS) seasons. We choose to plot these values against the mean SST of the Southern Ocean (Fig. 3), to show the potential relationships between the model-data agreement computed for each simulation and a cold or

220 warm Southern Ocean. We also choose to compute individual RMSEs for each ocean basin according to the core locations of the MARGO data, separating the Southern Ocean into two sectors (Atlantic and Indian sectors versus Pacific sector). The poorest agreement is observed in the Southern Ocean, especially in the Atlantic and Indian sectors of the Southern Ocean. The simulations with a colder Southern Ocean ('Cold P2', 'P4-I brines') show a better agreement with the SST data, as indicated by a smaller RMSEs computed for the Southern Ocean (see triangles in Fig. 3). However, 'Cold P2' is not the simulation with the

225 lowest mean RMSE (see crosses in Fig. 3b), as it notably shows a higher RMSE in the Atlantic basin in winter (see diamonds).

To better understand the discrepancies between data and model, we analyse next the SSTs in a data versus model diagram for the summer and winter months with superimposed information about their latitudinal location. A set of representative simulations are presented in Fig. 4, the interested reader can find similar plots for all simulations in Fig. S2. In general, the simulated LGM SSTs in austral winter (Fig. 4d,f,h) agree reasonably well with MARGO data. Although data are scarce in the

230 Southern Ocean for these winter months, it seems that simulations with a cold Southern Ocean ('P4-I brines') yield a better agreement with data (compared to 'P4-I' or 'P4-I hosing'). However, during the austral summer months, a clear trend with latitude is observed for all LGM simulations (Fig. 4c,e,g), with the model-data disagreement peaking around 40–50°S. At these latitudes, the summer Southern Ocean is too warm to match the data, even when taking into account the uncertainties. We note that the simulated summer SSTs in the Pacific sector of the Southern Ocean seem less overestimated (compared to

235 data) than in the Atlantic or Indian sectors. At higher latitudes (∼60°S), the agreement with data improves (as shown by points closer to the 1:1 line), and cold simulations even simulate colder summer SSTs than the SST data in the high latitudes of the Pacific sector, which is where sea ice is also simulated (see white markers in Fig. 4e and S2c, or Fig. S1c). This trend with latitude is almost as clear for the pre-industrial (Fig. 4a), which simulates a slightly too warm Southern Ocean compared to WOA98 data for most latitudes of the Southern Hemisphere, and for both seasons – though the model-data disagreement is

240 more pronounced in the summer months.

There is a clear anti-correlation between the simulated sea-surface temperature and sea-ice area in the Southern Ocean (Fig. S3), which suggests a thermodynamic control prevailing over the influence of advection processes. Therefore, we can also use sea-ice proxy data to further constrain the surface conditions, and examine whether our model-data evaluation using the sea-ice signal is consistent with our observations so far.

**3.3 Sea ice**

Analyzing correctly the sea-ice distribution requires distinguishing the summer and winter values. We here compare the simulated sea ice with data reconstructions for the austral summer (JFM) and winter (JAS) seasons, first in terms of sea-ice extent and then in terms of regional patterns. Only the sea-ice extent, defined as the surface with a sea-ice concentration over 15%, is strictly comparable to our data estimates. We however chose to present both the simulated sea-ice extent (here, the total surface

between the northernmost 15% concentration limit and the Antarctic continent) and area (the sea-ice concentration multiplied by the area of the grid cell for all ocean cells south of the equator) in Fig. 5.

Using the method described in Sect. 2.4 to integrate the sea-ice proxy data, we estimated a minimal (in austral summer) sea-ice extent of $\sim 10.2 \times 10^6$ km$^2$ and a maximal (in austral winter) extent of $\sim 32.9 \times 10^6$ km$^2$. This last value is significantly lower than previous studies ($39 \times 10^6$ km$^2$ in Gersonde et al. (2005) and $43.5 \times 10^6$ km$^2$ in Roche et al. (2012). While our estimates inherit the uncertainties linked to proxy data and to the extrapolation of sea-ice edges, this computation does not rely on a specific projection on a map. Given the limited change in the area enclosed in the contours, we estimate that the value of $43.5 \times 10^6$ km$^2$ of Roche et al. (2012) (which was also used in Marzocchi and Jansen (2017) to evaluate the simulated sea-ice extent of PMIP3 models) was overestimated. It is difficult to pinpoint the exact cause of this overestimation, but two factors certainly had a significant impact: first the use of a stereographic projection for the areal estimation, and second the use of the modern surface area of the Antarctic continent instead of the LGM one.

Comparing now these data reconstructions with our model outputs, Fig. 5 and S3a show that most simulations overestimate the LGM summer sea-ice extent – a tendency which is also noticeable for pre-industrial conditions (Fig. 5), despite the warm bias observed in Fig. 4a. Conversely, the sea-ice extent of most simulations fall close to the reconstructed winter sea-ice extent of $32.9 \times 10^6$ km$^2$. The warmest simulation ('P4-I hosing', see Fig. 1) is the only one to show both a winter and a summer sea-ice extent under the data estimates. However, simulations which are closer to the -4°C anomaly estimate (such as 'Warm P2' and 'New P2') show an overestimated minimal extent, yet a reasonable maximal extent, while warm simulations which are almost out of the $-4 \pm 0.8$°C range (such as 'P4-G' and 'P4-I') show both a small underestimation in winter and a small overestimation in summer. This suggests that the enhanced seasonality of the LGM Southern Ocean sea ice ($22.7 \times 10^6$ km$^2$ according to our proxy reconstructions, compared to the modern seasonal range of $15.4 \times 10^6$ km$^2$) is not entirely simulated by the model, a result already observed in Roche et al. (2012). Two sensitivity tests show opposite results: 'P4-I brines' shows a larger seasonality ($21.3 \times 10^6$ km$^2$) and 'P4-I wind' ($14.9 \times 10^6$ km$^2$) a reduced one compared to their parent simulation 'P4-I' ($16.7 \times 10^6$ km$^2$). It should be noted that, if we compared the simulated sea-ice area (instead of the extent) to our data estimates, we would rather conclude of a reasonable estimation of the sea-ice cover in summer for most simulations and of an almost systematic underestimation in winter. Indeed, the simulated sea-ice areas fall under the sea-ice extent values by 5 millions of square kilometers approximately, a difference enhanced in 'P4-I wind' due to the multiplication of the wind stress on ice.

Figure 6 presents the simulated sea-ice edges alongside the sea-ice contours based on marine core data, using the reconstruction method described in Sect. 2.4. The sea-ice edge – set at 15% of sea-ice concentration by convention (US National Snow and Ice Data Center) – of all LGM simulations shows a roughly circular regional distribution around Antarctica (also see Fig. S4). While the scarcity of summer LGM sea-ice indicators does not allow to make firm statements for the minimum extent, the circular shape does not compare well with the more oval-shaped proxy reconstruction in winter (Fig. 6b). Indeed, while cold simulations seem close to the reconstruction in the Atlantic and Indian sectors, they overestimate sea ice in the Pacific sector compared to proxy data. In summer (Fig. 6a), we observe a similar trend with less available proxy data: the simulated sea ice seems too extensive in the Pacific sector for cold simulations, but can not match some of the sea-ice presence indications in marine cores (reaching as far as 50°S in a few cores of the Atlantic sector). As the high southern latitudes of the Pacific are also where the model tends to simulate colder SSTs than MARGO data – on the contrary to the warm bias around latitudes of

40–50°S in the Atlantic and Indian sectors (Fig. 4 and S1), the observed discrepancies in sea-ice distribution seem consistent with the SST signal.

Both the SST and the sea-ice model-data comparison suggest that a cold Southern Ocean, with an relatively extensive winter sea-ice cover (which is present in some of our simulations), but also with both a large seasonal amplitude (simulated to a certain extent by one of our simulations) and a large interbasin contrast (shown by none of our simulations), would agree best with proxy data. Now that we have clarified what an improvement of the simulated surface conditions with respect to proxy data means, we can further use their variety to examine whether improved surface conditions would be linked to a more realistic water mass distribution.

## 3.4 Deep ocean circulation

Although all of our simulations broadly show the same biases in the seasonal and regional patterns of the Southern Ocean surface conditions, they simulate a variety of SST and sea-ice extent. We can expect these differences to have an impact on the density of surface waters and possibly on deep water formation. Additionally, since these surface conditions are simulated using different boundary conditions and/or forcings or model parameter choices (in the sensitivity tests), we take this opportunity to investigate the relative impact of these modelling choices and boundary conditions on the simulated deep ocean circulation.

We can examine the impact of the different modelling choices on the streamfunction along a meridional section of the Atlantic and Southern Ocean basins (Fig. 7). The AMOC depth and strength in our PI simulation are within the PMIP3/PMIP4 ensemble (see Fig. S1 and S2 of Kageyama et al. (accepted, 2021)). In more details, the streamfunction of iLOVECLIM is fairly comparable to the pre-industrial streamfunctions of HadCM3, AWIESM2, MIROC-ESM and CNRM-CM5, and actually stronger and deeper than that of IPSL-CM5A2 (and IPSL-CM5A-LR). However, the pre-industrial AMOC strength simulated by the iLOVECLIM model is underestimated compared to modern observational data. Since 2004, the RAPID array at 26°N has measured an AMOC within the range of 13.5 Sv to 20.9 Sv, when interannual variability is accounted for (Moat et al., 2020), with a mean estimate of 17.2 Sv (McCarthy et al., 2015). The simulated AMOC strength at this latitude does not fall into this range in any of our PI simulations, which show a maximum of 10.1 Sv ('PI') and 11.2 Sv ('PI brines', Fig. S5), with both maxima occurring at depth 1225 m. A clockwise cell can be observed in the Atlantic, which relates to the formation of NADW. In the Southern Ocean, we choose to define two anticlockwise cells: one which is located around 60–80°S, and another which is located both deeper and further north – but which do not always penetrate into the Atlantic Ocean. We name these three overturning cells the NADW cell, the Southern Ocean cell and the bottom cell respectively. As Otto-Bliesner et al. (2007) have shown, iLOVECLIM is among the models which simulate a very strong glacial NADW cell at the expense of the bottom cell (as is also the case here for almost all experimental settings, see Fig. 7b,c,d,e,f,h,i), a response which is not consistent with the shallower glacial NADW and the more voluminous AABW inferred from paleotracer data (Curry and Oppo, 2005; Howe et al., 2016; Böhm et al., 2015; Lynch-Stiglitz et al., 2007).

We first observe an effect of the boundary conditions choice. For example, the use of the new bathymetry generation method reduces the LGM NADW cell slightly: its overturning is more intense for 'Warm P2' than for 'New P2'. We also notice differences between the 'P4-G' and 'P4-I' streamfunctions, with a slight enhancement of the bottom overturning cell in the 'P4-G'

simulation associated with GLAC-1D (compared to the 'P4-I' simulation with ICE-6G-C), but not enough to counterweight the massive NADW cell. However, we note that the choice of forcings and model parameters seems to have a stronger impact than the boundary conditions, as evidenced by the contrasting results between the three sensitivity tests and their parent simulation 'P4-I'. The bottom cell is strongly enhanced by the use of the parameterization of the sinking brines, an experimental setting

which allows for the penetration of AABW in the Atlantic. On the other hand, the Southern Ocean cell is enhanced for 'P4-I wind', but moderately ('P4-I hosing') or strongly ('P4-I brines') suppressed for the other sensitivity tests. These results could be due to the fact that the experimental setting of 'P4-I wind' – with the multiplication of the meridional wind stress on ice – enhances sea-ice export, which leads to an increased sea-ice formation and its consequent brine rejection (Shin et al., 2003). In 'P4-I brines', the Southern Ocean overturning is not fully explicitly computed due to the parameterization, leading to these

very low values. Finally, it is no surprise that the addition of a freshwater flux ('P4-I hosing') leads to less overturning as it decreases the density of surface waters.

To single out the impact of surface conditions on the convection, we plot the relationship between the mean SST in the Southern Ocean and the maximum intensity of the three overturning cells in Fig. 8, for all simulations except the two with modelling choices affecting the density processes ('P4-I brines' and 'P4-I hosing', plotted on Fig. S7). The correlation coeffi-

335 cients R are very significant (with $|R| \geq 0.83$ for all plots), showing that simulations with a colder Southern Ocean tend to be associated with a stronger Southern Ocean cell, a weaker bottom cell and a more intense NADW cell. While this relationship holds, modelling choices yielding colder SST in the Southern Ocean (thus in better agreement with the data) do not lead to more realistic water mass distributions. Instead, a Southern Ocean cooling seems associated with an intensification of the open ocean convection, with a negative effect on stratification.

## 4    Discussion

### 4.1    What is the relative impact of boundary conditions and modelling choices?

With this set of simulations, we make use of the recent evolution of the iLOVECLIM model (regarding the recommended PMIP4 experimental design and its implementation, see Sect. 2.2) to investigate the relative impact of boundary conditions and of other modelling choices (related to forcings or model parameter choices) on the simulated surface conditions and deep

ocean circulation. Given the uncertainties in the ice sheet reconstructions, Kageyama et al. (2017) gave several options to modelling groups in the current phase 4 of PMIP, and advised the use of the new ICE-6G-C and GLAC-1D topographies (either one or, ideally, both). We have implemented both topographies in the relatively coarse resolution iLOVECLIM model and we show here that these two boundary conditions yield only small differences on the variables observed in this study. The use of the PMIP2 (ICE-5G) ice sheet reconstruction – with a higher elevation – causes an overall colder climate compared

to PMIP4 but differences in simulated surface conditions and deep ocean circulation remain relatively small. In contrast, the modelling choices made in sensitivity tests can cause much larger differences (e.g. between 'Cold P2' and 'Warm P2', or 'P4-I' and 'P4-I brines', or 'P4-I' and 'P4-I hosing'). In particular, the differences between 'Cold P2' and 'Warm P2' suggest that, while iLOVECLIM generally simulates a more modest global SAT anomaly than other PMIP4 models (Kageyama et al.,

accepted, 2021), modelling choices related to the glacial temperature profiles used in the radiative code can induce a very significant change. Moreover, thanks to the use of proxy data to evaluate our simulations, this inverse methodology approach is useful to highlight systematic biases in the simulated surface conditions of the Southern Ocean. In the iLOVECLIM model, it seems that the recurrent biases are larger than the differences related to the choice of boundary conditions. It is therefore particularly important to investigate and understand the origin of these biases, while different ice sheet reconstructions have a relatively smaller impact and may not all be implemented during the PMIP4 exercise. Nonetheless, it should be noted that Galbraith and de Lavergne (2019) have investigated the effects of a broader range of forcings (greenhouse gas concentrations and orbital parameters in addition to changes in ice sheet size) on the deep water masses and they notably highlighted the nonlinear responses of their volume to varying forcings (e.g. with different global temperatures). Therefore, the choice of ice sheet reconstruction could potentially yield more significant differences in deep ocean circulation under different time periods or simulated global temperature.

## 4.2 What is the "best" simulation, and why?

Our analysis suggests that in terms of surface conditions, the PMIP2 boundary conditions yield a better agreement than the PMIP4 ones with SST and sea-ice geological data. However, among our set of eight simulations, the sensitivity test with the parameterization of the sinking of the dense water ('P4-I brines') is the one with the best overall agreement with data. This parameterization allows for the simulation of a cold Southern Ocean, an extensive winter sea-ice cover along with an enhanced seasonality of sea ice (close to the data estimate) compared to other simulations. This parameterization also impacts the AABW density and, therefore, the deep ocean circulation. Among our set of simulations, it is the only one simulating a water mass distribution which is reconcilable with reconstructions from paleoproxies. Nonetheless, this experimental design (like all the others tested in this study) does not result in a shoaling of the AMOC between the PI and LGM state (see Fig. S5), as inferred from the majority of the proxy data (Curry and Oppo, 2005; Böhm et al., 2015; Skinner et al., 2017; Gebbie, 2014). In contrast, Morée et al. (2021) were able to simulate with the NorESM-OC model a shoaled and slightly weaker AMOC at the LGM compared to their PI state. As the radiocarbon ages simulated in southern source waters were too young compared to data, they however suggested that the ventilation at the LGM was still overestimated, possibly in relation to a too small Antarctic sea-ice extent in their LGM simulation (their Fig. S12, displaying a sea-ice extent of $\sim 4.94 \times 10^6$ km$^2$ in summer and $\sim 32.95 \times 10^6$ km$^2$ in winter). However, if we consider our new estimates of $\sim 10.2 \times 10^6$ km$^2$ and $\sim 32.9 \times 10^6$ km$^2$ (respectively for the summer and winter sea-ice extent inferred from proxy data), instead of the ones presented in Roche et al. (2012), the sea-ice extent simulated by Morée et al. (2021) is underestimated only in summer. Therefore additional processes might be involved to explain the weak ventilation of Southern Ocean sourced deep water at the LGM.

Artificially sinking dense waters is motivated by the fact that, due to the coarse resolution of the model, the salt linked to brine rejection during sea-ice formation tends to get diluted in the surface grid cells rather than allowing the sinking of dense water along the continental slope (Bouttes et al., 2010). Though legitimate, this parameterization is quite crude: a fraction (here chosen at 0.8) of the salt content of the surface grid cells is directly transferred to the deepest grid cell beneath them, without explicitly computing the convection.

However, we can argue that the open ocean convection in the Southern Ocean is actually hindering the simulation of a realistic water mass distribution. Indeed, while paleotracer data suggest a dense, stratified glacial deep ocean, the simulation of cold conditions in the Southern Ocean is rather associated with an intense convection in the Southern Ocean – therefore well-mixed, and a deep NADW (Fig. 7 and 8). As underlined by Heuzé et al. (2013), models struggle to simulate the correct bottom water properties even in the present-day conditions, as they tend to form AABW by open ocean convection, a process rarely observed, instead of the overflow of dense continental shelf water. While none of the CMIP5 models were able to simulate the latter, Heuzé (2021) showed that a few CMIP6 models are now able to simulate AABW formation via shelf processes, notably thanks to the development of an overflow parameterization. Despite this progress, the issue remains, as "the large majority of climate models form deep water via open ocean deep convection, too deep, too often, over too large an area" (Heuzé, 2021).

Our results suggest that, even if we were able to simulate surface conditions in perfect agreement with proxy data, it would probably not be sufficient to simulate a deep ocean circulation in good agreement with paleotracer data, unless the convection and mixing processes are realistically represented by the model. Accounting for the sinking of brines rejected during sea-ice formation using a parameterization may be one way of tackling this issue, but other authors have also put forward the importance of a realistic vertical mixing scheme (De Boer and Hogg, 2014; de Lavergne et al., 2017). Topography-dependent mixing parameterizations, linked to the energy received by water masses due to geothermal fluxes and interactions of tidal waves with the ocean floor, have been recently developed in some high resolution models (de Lavergne et al., 2019). Their effects on the simulated deep ocean circulation in a coarser resolution z-level model such as iLOVECLIM may be of interest for further studies.

## 4.3   What are the systematic biases?

Still, even in the sensitivity test with the parameterization of the sinking of the dense water ('P4-I brines'), which yields the best model-data agreement among our set of simulations, the model results show the same biases than in all the other simulations. We observe several systematic biases, linked to seasonal or regional patterns of SSTs and sea ice. First of all, the simulated seasonal amplitude of sea ice is too small with respect to the proxy data estimates, which suggest a sea-ice seasonality of $22.7 \times 10^6$ km$^2$ ($\pm 8.0 \times 10^6$ km$^2$ based on 15% and 30% error bars on winter and summer sea-ice extent, respectively). Secondly, the simulated winter sea-ice extent seems too small (compared to data) in the Atlantic and Indian sectors ($\sim$40–50°S), and too large in the Pacific sector ($\sim$60°S) for cold simulations. The model simulates round sea-ice distributions while proxy data suggest more oval-shaped winter and summer covers, as observed today. Thirdly, the simulated summer SSTs are too high in the Atlantic and Indian sectors ($\sim$40–50°S) with respect to MARGO data, while they sometimes seem slightly too low in the high latitudes of the Pacific sector. This is true at least for the summer months, as data are scarce in the winter months.

We note that the model underestimates the interbasin contrasts, as it struggles to simulate a large winter sea-ice cover in the Atlantic and Indian sectors. While a good representation of sea-ice advection by the Antarctic Circumpolar Current may be hard to achieve in key areas (e.g. Weddell Sea, Ross Sea, Kerguelen plateau where strong oceanic gyres exist) due to the discretisation of the coasts on a $3° \times 3°$ land-sea mask, this difficulty could be largely attributed to the warm bias

observed in the Southern Ocean. The clear zonal trend of this bias may stem from an underestimated polar amplification, and/or of the SST gradients across the oceanic fronts (whose location may also be wrong). This type of bias is not surprising considering the relatively coarse spatial resolution of iLOVECLIM. Interestingly, diverse modelling studies have also pointed out distinctive regional patterns in the Southern Ocean with significant differences between the Pacific sector and the other two sectors, whether it considered freshwater fluxes linked to icebergs and their influence on sea ice (Merino et al., 2016), zonal asymmetries of the Southern Hemisphere westerly jet trends (Waugh et al., in review, 2020), or the sea-ice retreat scenario with the best agreement with the Antarctic ice core $\delta^{18}$O records at the Last Interglacial (Holloway et al., 2017).

Identifying the origin of a bias is always a challenge. It might be an especially hard task to identify the origin of biases in the simulated sea-ice cover, considering the sheer number of feedbacks involved (Goosse et al., 2018). What can be noticed is that the simulated sea-ice seasonal cycle is affected by some of our modelling choices (increased in 'P4-I brines', reduced in 'P4-I wind'). Alongside, the Southern Ocean convection is suppressed in the first sensitivity test, and enhanced in the second. In a climatological mean in our model there seems to be a link between reduced Southern Ocean convection and increased sea-ice seasonal cycle. In opposition to this observation, Heuzé et al. (2013) have underlined the fact that CMIP5 models with a large sea-ice seasonality are also the ones simulating open ocean convection over extensive areas at modern times, arguing that strong sea-ice formation could precondition the ocean for open ocean deep convection. This questions the relative importance of the different simulated mechanisms at play linking the ocean convection and the sea-ice seasonal cycle, an aspect that is present in several studies (Marshall and Speer, 2012; Behrens et al., 2016; Ma et al., 2020).

## 5   Conclusions

Using diverse boundary conditions and sensitivity tests, we are able to simulate a variety of LGM climates, and in particular different surface conditions in the Southern Ocean among our set of simulations. We assess the model-data agreement in terms of both SSTs and sea-ice extent, and explore the associated impact on deep ocean circulation.

In this study, we underline that simulated cold surface conditions in the Southern Ocean are overall in better agreement with proxy data. A detailed analysis shows that there are seasonal and spatial distribution patterns which are associated with systematic discrepancies between our simulations and both sea ice and SST reconstructions. All simulations underestimate the sea-ice seasonal range (with a simulated sea-ice extent range equal to 65% to 94% of the range inferred from the proxy reconstructions). Model-data comparisons also consistently suggest that the simulated SSTs of the Pacific sector of the Southern Ocean (∼60°S) are slightly too low while those of the Atlantic and Indian sectors (∼40–50°S) are too high, which may explain why the model is not able to reproduce the reconstructed oval-shaped distributions of sea ice. Overall, the model results exhibit a mean warm bias of 2 to 6°C over the Southern Ocean with respect to MARGO data.

Yet, colder conditions in the Southern Ocean would not necessarily lead to a more realistic water mass distribution. Our study shows that colder conditions rather tend to intensify the Southern Ocean open ocean convection, a process which leads to inaccurate AABW properties, as it does not account for the overflow of dense continental shelf water but instead creates a well-mixed water column. The parameterization of the sinking of brines is the only experimental setting we used which

accommodates a better representation of both the surface conditions and the deep ocean circulation. For the variables analyzed in this study, it would therefore seem that the improved simulation of convection processes is paramount, and far more important than the choice of ice sheet reconstruction used to implement the orography and bathymetry.

*Data availability.* The model outputs and reconstructed sea-ice limits are available for download online (doi: 10.5281/zenodo.4576026).

## Appendix A: Description of the semi-automated method to generate CLIO bathymetries

This method replaces the tedious manual changes that have been done on the CLIO grid in the past, in order to be able to generate a CLIO bathymetry quickly from any topography file – a technical development which fastens the start of new PMIP phases and enables the run of transient simulations with an interactive bathymetry. It has been used here to (re)generate a pre-industrial bathymetry (using the high resolution etopo1 topography), a PMIP2 bathymetry (using the ICE-5G reconstruction), and two PMIP4 bathymetries (using either the GLAC-1D or the ICE-6G-C reconstruction).

This development has been done in several pre-processing steps :

– Anomalies are computed using the PMIP2/PMIP4 topographies and then regridded on the etopo1 grid :

$$\text{LGM topography} = \text{PI (etopo1)} + \text{LGM Peltier (ICE-6G-C, 21kyr)} - \text{PI Peltier (ICE-6G-C, 0kyr)}$$

– A connectivity program (see Appendix B) writes the mean bathymetry and hypsometry into a text file, either on the
rotated or regular CLIO grid. It also produces the connections between ocean basins thanks to the computation of subgrid sills.

– In a second program, the two grids are first put together.

– Then, the mask is generated using the hypsometry, a chosen sea-level (-0.5 m for the PI, -133.9 m for the LGM, according to Lambeck et al. (2014)), and a chosen threshold (% of surface of a grid cell above which the cell is defined as ocean
- here 40 %). Small isolated seas are closed. The mask of a few ocean grid cells is manually forced at the PI so that all the critical straits stay open. These manual points have to be redefined at the LGM. Indeed, while some stay the same (Gibraltar), others are not necessary anymore (Hudson Bay and Japan Sea outlets) and a few new critical points appear (Fram Strait, Golf of Mexico outlet). We take particular care of this step, using the connections computed earlier and our knowledge of the LGM ocean.

– The bathymetry is converted into the irregular vertical levels of the CLIO model. The new vertical levels are set equal to the former vertical levels for a few problematic grid cells in order to get realistic salinity values in the Mediterranean Sea and Hudson Bay. The vertical level 1 is avoided (either forced to 0 or 2), because the model cannot deal with these very

shallow grid cells. As the model also cannot deal with isolated oceanic grid cells for which the deepest vertical level is isolated (e.g. deep grid cells with shallower neighbours), a process similar to a smoothing filter is applied.

– Finally, this program writes a text file containing the bathymetry with the land-sea mask (0 in every land grid cells).

– Two additional pre-processing steps are required to generate the necessary input files (one containing the fraction of ocean seen by the T21 grid cells, another containing the interpolation points between the CLIO and the T21 grids).

– In order to be able to quickly equilibrate the model when running a simulation with a different bathymetry than its restart, the initialization code of iLOVECLIM has been modified to generate realistic values of the tracers content of new oceanic 490 grid cells. To achieve this, the initialization of all the restart variables in new ocean grid cells is done by averaging the values in neighbouring oceanic grid cells when necessary. The conservation of the total content of conservative variables (salt, carbon...) is ensured.

## Appendix B: Description of the connectivity program

The software "topo_connect" was developed in order to compute the connection between ocean basins directly from topogra-495 phy/bathymetry data. The basic idea is rather simple, though its implementation is not trivial. The algorithm builds a global tree structure from the topographic data file, with each leaf corresponding to a local minimum in the topographic data, with the trunk corresponding to the entire domain, and with branching occurring for each sill between two (or more) sub-basins. From this tree structure, it is then easy to find the lowest sill connecting any two points, by finding the first common branch to which they belong.

More precisely, the algorithm starts by finding the local minima in the topographic domain. For each minimum, it builds the set of points belonging to this minimum basin by adding the lowest (uphill) neighbour, and continues to do so up to finding a sill. This sill corresponds to a branching between two (or more) basins. The algorithm then continues the same procedure from this sill, up to the next one, and so on, until all basins (branches) are connected to a single trunk, which represents the whole domain. Building this tree structure is the most computationally demanding task. Then, for any two points in the domain, it 505 is easy to use this tree structure and to find the level (and the location) of the lowest sill connecting them. For a given sea level, this allows to decide if two ocean basins are connected or not, according to topographic data. This information is then aggregated in a new grid system, typically an ocean model grid with a much lower resolution, in order to decide whether model cells are connected or not.

The implementation requires caution, since non-trivial cases can arise. For example, there may be flat areas in the domain 510 and/or multiple sills at the same level therefore connecting more than two basins at the same time. The implementation relies therefore not on simple traditional "arrays" but on more flexible structures like "lists" or "priority_queues" available in standard C++. The execution time is a few minutes on a desktop computer when using bathymetric data at the resolution of 1 arc-minute (etopo1). Higher resolution could be useful to resolve some canyons, in particular to compute the possible extent of lakes on land, but this was not investigated so far.

*Author contributions.* FL, NB and DMR designed the study. DP developed the connectivity program. DMR, FL and NB developed the semi-automated bathymetry generation method, and the iLOVECLIM model to accomodate new boundary conditions. FL performed the simulations and analyzed the outputs under supervision of NB and DMR. XC compiled existing sea-ice proxy data, inferred the reconstruction of LGM sea-ice edges and provided expert knowledge on sea-ice processes. CW contributed expert knowledge on SST data. FL wrote the manuscript with contributions from all co-authors.

*Competing interests.* The authors declare that they have no conflict of interest.

*Acknowledgements.* This work was supported by the French National program LEFE (*Les Enveloppes Fluides et l'Environnement*). FL acknowledges the use of the LSCE storage and computing facilities. DMR, NB, CW and XC are supported by the *Centre national de la recherche scientifique* (CNRS). DMR is also supported by the Vrije Universiteit Amsterdam. DP is supported by the *Commisariat à l'énergie atomatique et aux énergies alternatives* (CEA) and FL by the Université Versailles Saint-Quentin-en-Yvelines (UVSQ). CW acknowledges support from the European Research Council ERC grant ACCLIMATE/n° 339108. We thank Aurélien Quiquet for scientific discussions and technical support on the use of iLOVECLIM. We also thank Masa Kageyama and Jean-Yves Peterschmitt for their technical help. Last but not least, we thank the two anonymous reviewers for their help with the manuscript.

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

**Table 1.** Short description of the iLOVECLIM simulations

| Simulation | Duration (years) | Forcing parameters | Topography | Bathymetry | Comments on the experimental setting |
|---|---|---|---|---|---|
| PI | 5000 | PI | default | semi-automated* | Reference simulation for the pre-industrial climate |
| Cold P2 | 5000 | PMIP2 | ICE-5G | manual | With the default profile, yielding a very cold climate |
| Warm P2 | 3000 | PMIP2 | ICE-5G | manual | With the Greenland profile allocated to all ice-covered regions in the Northern Hemisphere |
| New P2 | 5000 | PMIP2 | ICE-5G | semi-automated | Reference LGM simulation with boundary conditions associated with PMIP2 |
| P4-G | 5000 | PMIP4 | GLAC-1D | semi-automated | Reference LGM simulation with boundary conditions associated with GLAC-1D** |
| P4-I | 5000 | PMIP4 | ICE-6G-C | semi-automated | Reference LGM simulation with boundary conditions associated with ICE-6G-C** |
| P4-I brines | 5000 | PMIP4 | ICE-6G-C | semi-automated | Sensitivity test with the parameterization of the sinking of brines |
| P4-I wind | 3000 | PMIP4 | ICE-6G-C | semi-automated | Sensitivity test with the multiplication by 3 of the meridional wind tension on ice |
| P4-I hosing | 3000 | PMIP4 | ICE-6G-C | semi-automated | Sensitivity test with hosing (+0.6 Sv) around Antarctica |

*generated using etopo1 (Amante and Eakins, 2009)

**both simulations are also part of Kageyama et al. (accepted, 2021)

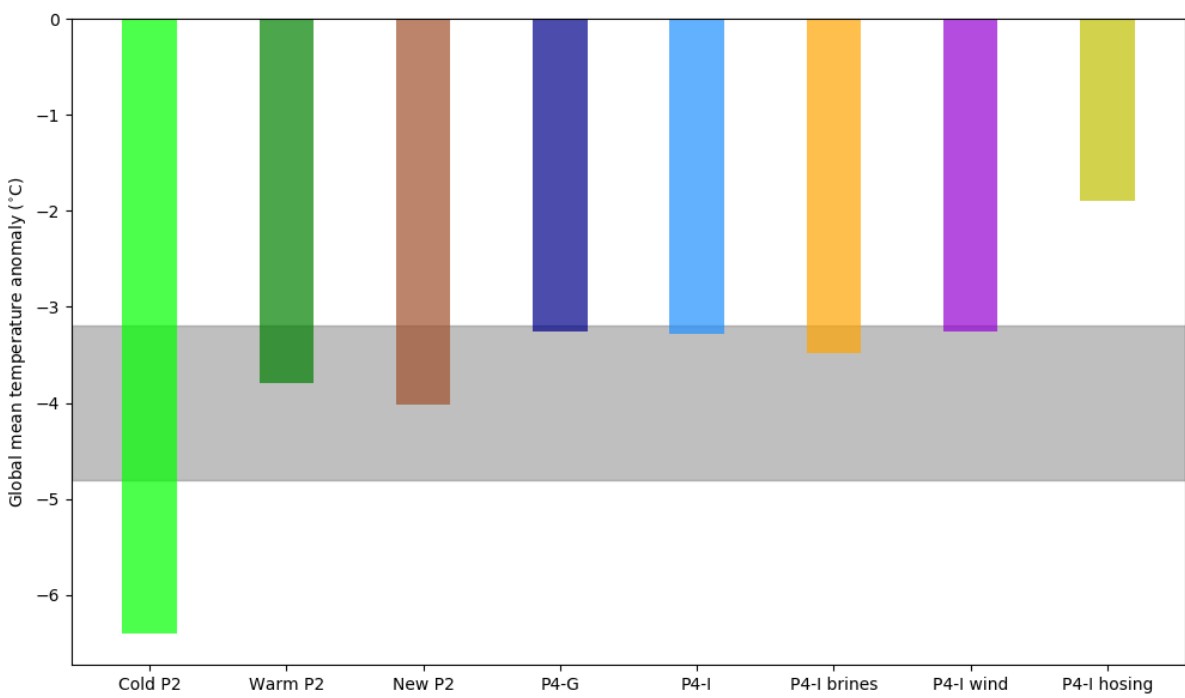

**Figure 1.** Global mean surface air temperature anomalies (LGM – PI). The grey bar shows the anomaly (-4 ± 0.8 °C) estimated by Annan and Hargreaves (2013).

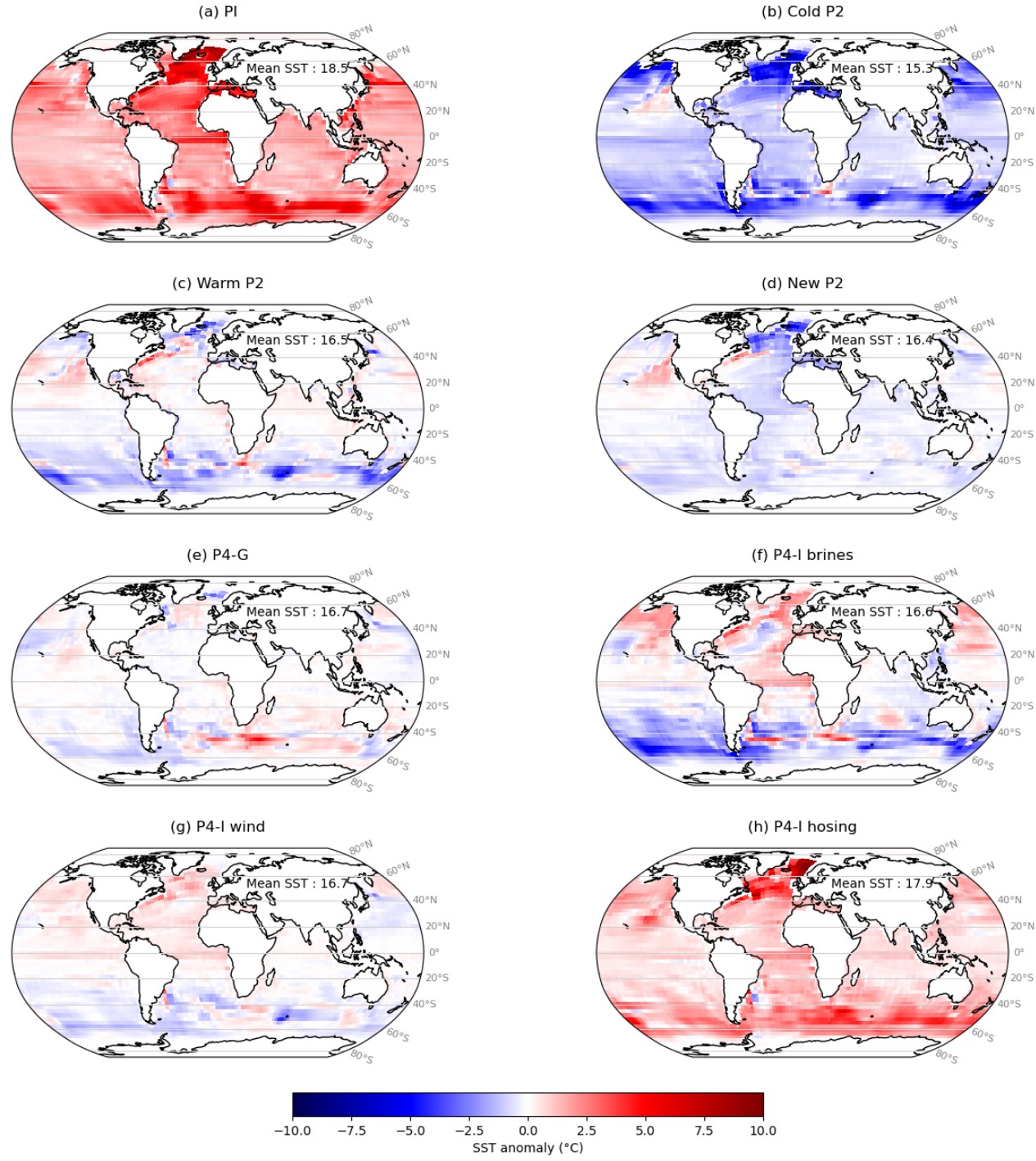

**Figure 2.** Anomaly in simulated mean sea-surface temperature (°C) relative to simulation 'P4-I' (mean SST = 15.2 °C). Due to the vertical resolution of iLOVECLIM, the sea-surface temperature is defined as the temperature of the first 10 m of the water column.

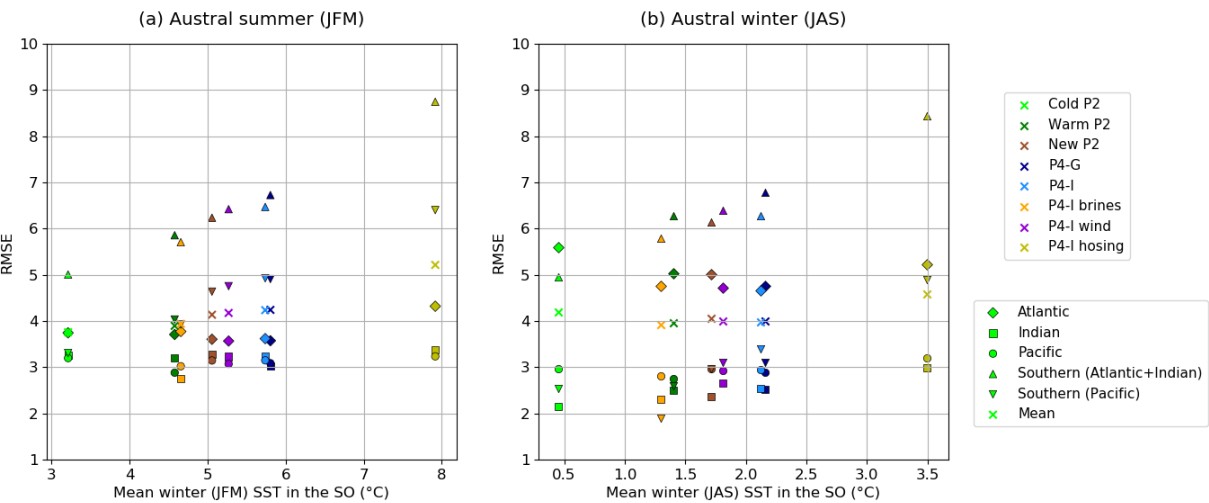

**Figure 3.** Relationship between the model-data agreement and the overall temperature of the Southern Ocean, in austral summer (a) and winter (b). The mean value of the Southern Ocean SSTs (averaged up to 36°S) of each simulation is plotted on the x-axis. The y-axis represents the root mean square error computed using the SST data from MARGO Project Members (2009), which is small when the agreement is good. This value was computed for each basin and each simulation, as shown by the marker style and color respectively.

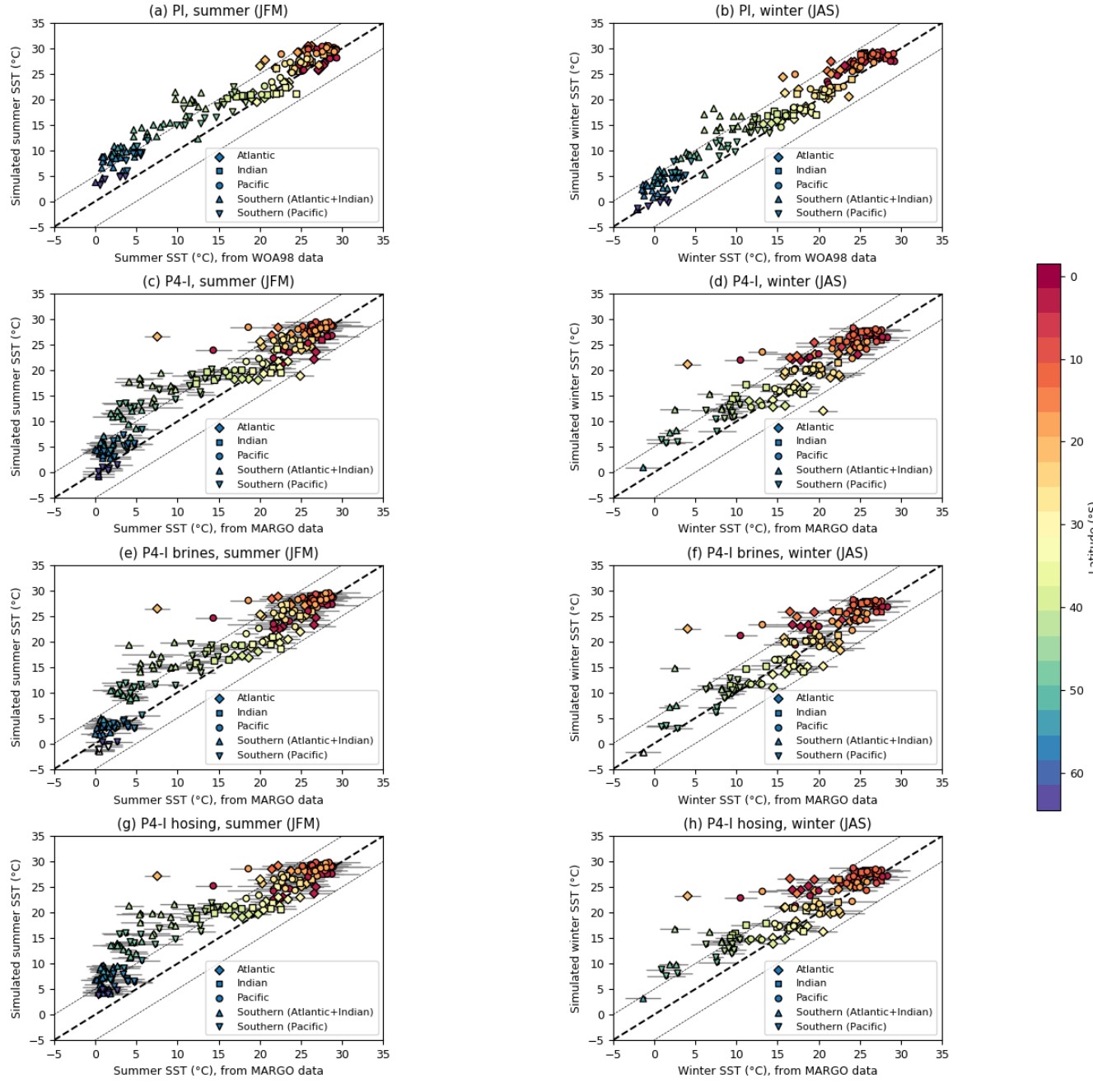

**Figure 4.** Austral summer (JFM) and winter (JAS) sea-surface temperatures of the Southern Hemisphere in a model versus data diagram, for a sample of our simulations (see Fig. S2 for the complete results). The simulated SSTs are plotted against the SST data from the regridded product (MARGO Project Members (2009) or World Ocean Atlas (1998)) thanks to the aggregation of the coordinates on the nearest ocean grid cell. The 1:1 line features a perfect model-data agreement (black dashed line), while the grey dotted lines features a 5°C departure from it. The marker style indicates the ocean basin of each core. The marker color shows the latitude of the core, except it is white where the model simulates sea ice in the Southern Ocean. The uncertainties associated with the SST data are plotted by the grey horizontal bars.

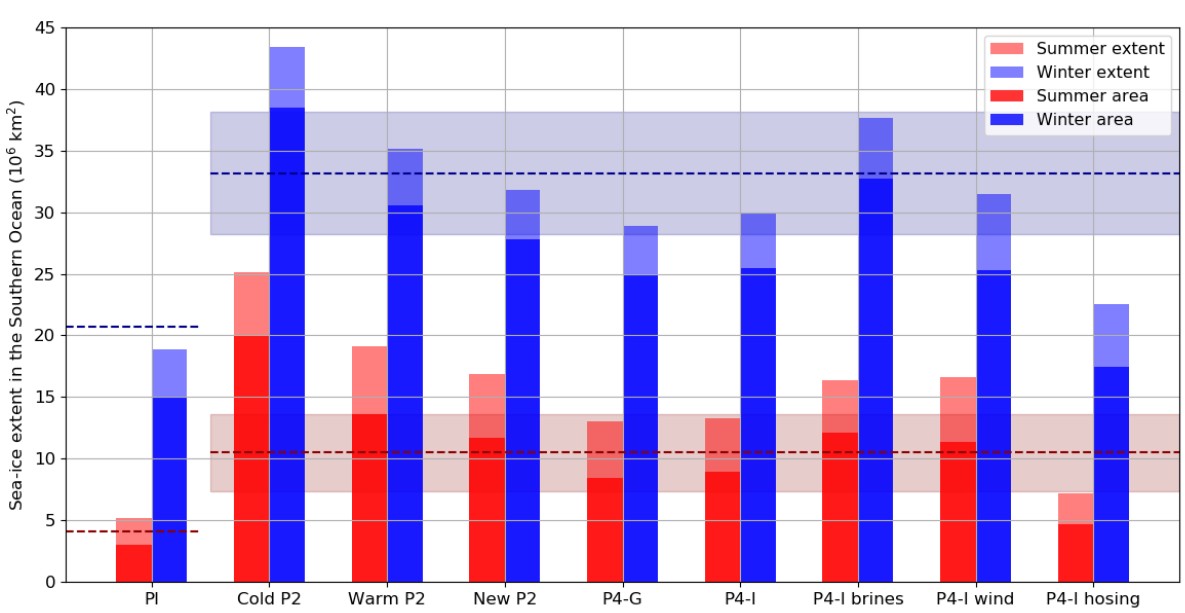

**Figure 5.** Austral summer (JFM) and winter (JAS) sea-ice areas and extents in the Southern Ocean. The LGM sea-ice extent estimated using the proxy data compilation is represented by the red (summer) and the blue (winter) dashed lines (with an indicative error bar of 30% and 15% respectively). The modern values (dashed lines on the left) are mean values on the period 1979–2010 published in Parkinson and Cavalieri (2012).

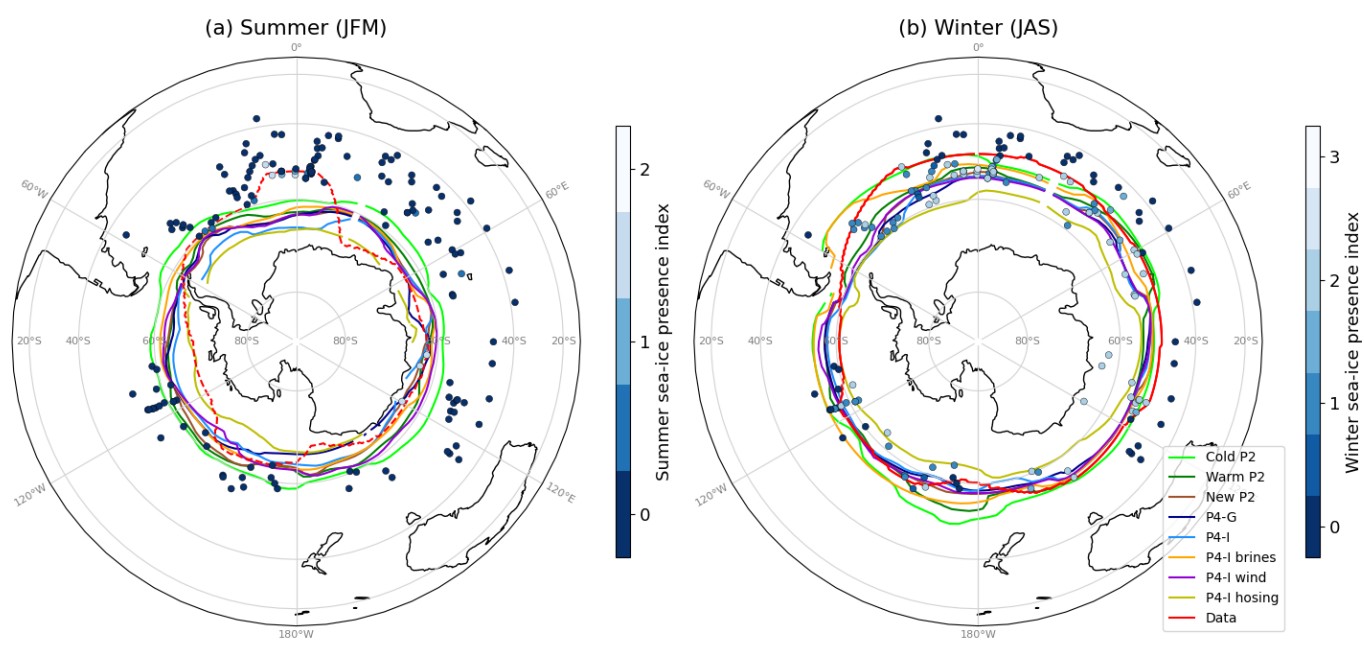

**Figure 6.** Austral summer (a) and winter (b) sea-ice edges (at 15% of sea-ice concentration, enclosing the total ocean surface defined as the sea-ice extent) in the Southern Ocean for the LGM simulations. The sea-ice presence suggested by marine cores data is represented as an arbitrary index on a blue to white scale, where blue denotes no indication of sea ice in proxies, and white denotes agreement of several proxies on the presence of sea ice. The red lines mark the likely delimitation of the sea-ice presence according to the proxy data (compilation of data from Gersonde et al. (2005), Allen et al. (2011), Ferry et al. (2015), Benz et al. (2016), Xiao et al. (2016), Nair et al. (2019), and Ghadi et al. (2020)). We used a solid red line in (b) but a dashed line in (a) as the summer contour is not well-constrained (see Sect. 2.4).

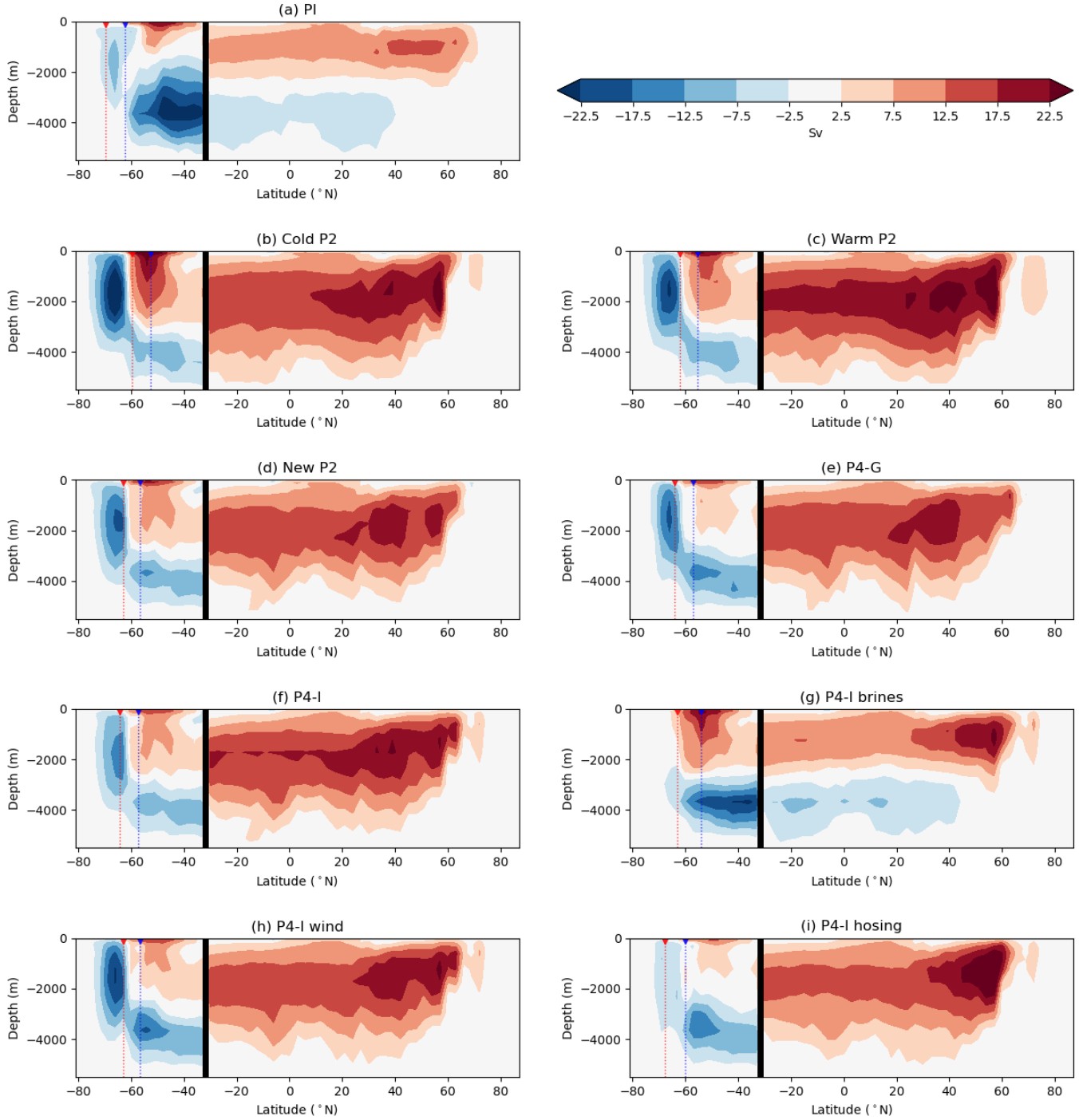

**Figure 7.** Streamfunctions (Sv) in the Atlantic (North of 32°S) and Southern Ocean basins (South of 32°S). The black vertical line represents the limit between these two basins, chosen at 32°S. The thin dotted lines show the latitude of the average sea-ice edge in austral summer (red) and winter (blue) for each simulation.

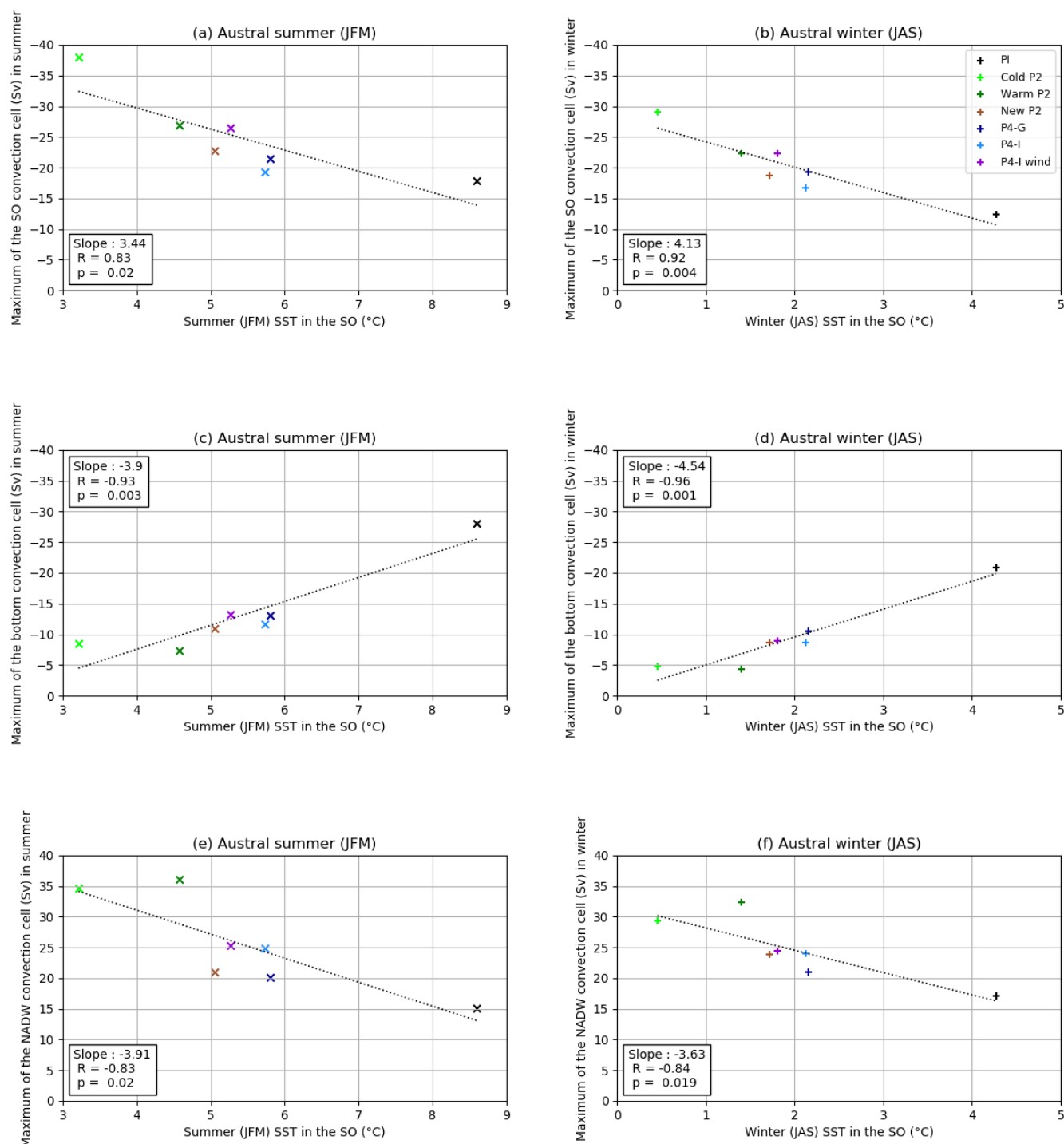

**Figure 8.** Relationships between the mean SST in the Southern Ocean (averaged up to 36°S) and the Southern Ocean (a, b), bottom (c, d) or NADW (e, f) overturning cell maximum, for all simulations except 'P4-I brines' and 'P4-I hosing'. The y-axis is inverted for the two anticlockwise cells (a, b, c, d). The dotted line represents the linear fit to the model results plotted here.