# Peer review of "Impact of Southern Ocean surface conditions on deep ocean circulation at the LGM: a model analysis"

_Climate of the Past, 2020_

## Referee Comment (RC1) · Anonymous Referee #1 · 19 Dec 2020

This paper considers a series of simulations using the iLOVECLIM model to analyze the effect of different choices of LGM boundary conditions, parameterizations, and ad-hoc modifications (to wind stress and freshwater forcing) on the LGM solution, focussing on the surface climate as well as the deep ocean overturning. Although the results are generally interesting and worthy of discussion, I think the comparison between these experiments and conclusions about which "choices" are better or more important then others need to be drawn a bit more carefully, as these various modifications aren't really comparable and some are certainly more physical then others. The paper specifically highlights the use of a parameterization to represent "sinking brines" as key to obtaining a realistic simulation of the LGM overturning circulation. However,

I believe that the discussion of these results needs improvement, as elaborated below. Most importantly, I think that a PI simulations with this parameterization needs to be presented for comparison. Since the physics of the ocean have not changed between the present and LGM, the model needs to be able to reproduce the PI ocean and LGM ocean circulation with the same parameterization.

Specific comments:

- l. 15/16 in abstract. It would be good to clarify that you are referring to "different choices for LGM boundary conditions", or better yet "different choices for the LGM ice sheet topography" (see also comment below). After all, differences in boundary conditions between the PI and LGM are ultimately what has to explain the different circulation in the two climates.

- In various places (e.g. l. 36-38) the issue of open ocean convection versus sinking along the AA slope seems to be used almost synonymously to the role of brine rejection in deep water formation, but these are rather different processes. Notably, the CCSM3 LGM simulation shows very salty AABW, clearly as a result of strong brine rejection, yet I assume AABW is still formed by open ocean convection (e.g. Shin et al. 2003, https://doi.org/10.1029/2002GL015513 - indeed this paper should be discussed).

- Relatedly, the parameterization of sinking brines needs to be described a little more– both in terms of the formulation and its physical interpretation. I understand that this method has been published previously, but since it is key to the presented conclusion I think the reader needs to be able to interpret the results from these simulations without first reading Bouttes et al. (2010). In l. 125 it is argued that "its objective is to account for the sinking of dense water along the Antarctic continental slope", and similarly in l. 301 it is refereed to as "the parameterization of the sinking of dense water along the continental slope". This gives the impression that it may be a parameterization of down-slopes gravity currents, which, however, seems quite misleading. Indeed, if I understand correctly, the parameterization simply transfers salt from brine rejection

directly and locally to the bottom of the ocean (without any mixing along the way), and it is not limited to the Antarctic slope. Personally, I'll have to admit that this parameterization seems rather unphysical to me (even gravity currents are associated with lots of entrainment and detrainment as they proceed down the slope, and of course they only exist on the slope). The readers can form their own opinion, but to do so, the parameterization needs to be discussed clearly.

- As discussed in the general comments above, the "P4-I brines" simulation needs to be compared to a corresponding PI simulation with the same parameterization. Ultimately the changes in the ocean circulation between the PI and LGM climates have to be attributable to differences in the boundary conditions, not different physics. It needs to be verified that the model is able to reproduce a reasonable solution for both the LGM and PI ocean with the same parameterzation. (One aspect that should be paid attention to here are the T and S properties of NADW and AABW. Importantly, AABW is fresher than NADW in the modern climate, which needs to be reproduced by any model that adequately represents watermass transformation processes in the SO.)

- in l. 131/132 it is argued that a quasi-equilibrium state is ensured. How is this evaluated?

- l.161: does "surface extent" here refer to sea ice extent or still the surface of the continent? I assume the former, but please clarify.

- The error bars of 10% and 20% for winter and summer sea ice extent seem to be mostly accounting for the uncertainty in the continental margins, which is probably relatively small compared to the large uncertainty in the sea ice line. As a result, these error estimates seem very optimistic to me. Indeed this seems to be confirmed by the fact that the previous estimates of Gersonde at al. (2005) and Roche et al. (2012) fall significantly outside of this error bar. I'm not arguing that either estimate is better or worse, but simply that a larger uncertainty has to be acknowledged. I think the uncertainty range should at least encapsulate the best estimates of these major

previous studies.

- l. 220: It is argued that "the transfer of brines leads to a cooling of the Southern Ocean". Notably, however, the cooling does not occur in the regions of AABW formation but further north. There also is pronounced warming (relative to P4-I) in the North Atlantic. Do you have an explanation for these results? And does the warmer North Atlantic play a role in explaining the relatively weak and shallow AMOC in this simulation? The focus here seems to be almost entirely on the Southern Ocean, but what's the effect of the brine parameterization in the North Atlantic?

- l. 214: what is the statement that "'Cold P2' is not the simulation with the best overall agreement" based on? From what is shown in the paper, it seems to at least show among the best agreement in terms of SSTs. (And believing the Tierney et al. (2020) estimate it would also be the best in terms of global mean temperature.)

- Fig. 7: What exactly is plotted here? Is it only the resolved Eulerian mean overturning or does this include the parameterized eddy transport associated with the GM parameterization? What matters for the transport of physical and geochemical tracers is really the isopycnal overturning (which probably does not have two counter-clockwise cells in SO). Computing the latter is admittedly more challenging and not commonly done in studies like this, but at least the GM contribution should be included.

- From Fig. 7, it also seems that the AMOC in the PI simulation is too weak and too shallow, which should be discussed.

- For the evaluation of deep ocean water mass properties it would be very useful to show T and S (as a function of latitude and depth).

- l. 303-305: I don't follow the argument here about why 'P4-I wind' has a stronger Southern Ocean cell. My guess would be that the stronger wind stress over ice leads to enhanced ice export, which in turn leads to more new sea ice formation and thus brine rejection (c.f. Shin et al 2003).

- In Fig. 8 and throughout much of the manuscript "convection" seems to be used synonymously to the large-scale overturning circulation. However, convection can occur without a large-scale overturning and vice versa. I suggest to replace all references to convection cells with overturning cels.

- In section 4.1 the various simulations are separated into those that amount to different choices for "boundary conditions" and "experimental setting", a separation that makes its way also into the abstract and conclusions. This separation, and the term "experimental setting" seems very vague. E.g. the assumed glacial temperature profile affects the simulation results via heat flux in or out of the glacier surface, and thus effectively also amounts to a difference in boundary conditions. In general it seems that "boundary conditions" is only used for cases with different choices for ice sheet topography, so I suggest to simply be explicit about that. As for the various other experiments, I don't see how they can be lumped into one category.

- Section 4.3.: Given the high uncertainty, particularly in the reconstructions of summer sea ice cover, I think it would be useful to provide some estimate of uncertainty for the sea ice seasonality from proxy data.

- I find the last paragraph of section 4.3 and specifically the attempt to reconcile the conflicting results between this study and Heuze et al. (2013) hard to follow and it seems very speculative. I don't think this discussion is necessary either, so I suggest removing this paragraph.

- Based on the issues pointed out above, I'm not sure the last sentence of the conclusion can be justified. At the least, "boundary conditions, such as the ice sheet reconstruction" should be reduced to just "the ice sheet reconstruction".

---

## Referee Comment (RC2) · Anonymous Referee #2 · 4 Jan 2021

In this paper, Lhardy et al. use an intermediate-complexity model to evaluate the response of the simulated glacial ocean circulation to Southern Ocean surface ocean temperature and sea-ice conditions. They achieve differences in these surface conditions by running simulations with various options for glacial climate boundary conditions from the Paleo Model Intercomparison Project (PMIP), and by simulations where they change wind (i.e. sea-ice export) conditions, formation of salty brines, and freshwater input in the Southern Ocean. They find that the sensitivity tests with winds, brines, and freshwater have more potential to influence the simulated surface properties, and particularly the ocean overturning circulation and distribution of water masses, compared to the choice of boundary conditions. This highlights the importance of informed

choices in model parameterizations of processes. In particular, it further clarifies the effect of deep-water formation and convection processes for achieving a realistic representation of the glacial deep ocean and its water masses. This is of importance for our ability to understand, and simulate, glacial ocean storage of $CO_2$, and to improve our models. As this category of models is commonly used for paleo simulations, this study is particularly educational for modelling groups using models of similar resolution and complexity, especially those participating in PMIP.

The experimental design is overall sound, though adding a PI state that uses the brines parameterization would be beneficial, to test the effect of 'a better representation of deep-water formation' in the modern ocean. As the parameterization does not change the amount of sea ice, only its effect on water mass properties and circulation, the choice of 0.8 as the scaling should not have to change between climate states. The amount of brines, and thus their influence on the ocean properties, should decrease in a warmer climate due to the reduction in sea-ice. It would also be useful to test the brines parameterization together with the PMIP2 boundary conditions, to strengthen some of the conclusions about the role of boundary conditions, but I leave it up to the authors to decide if this is feasible.

Overall, the paper is well written, with a generally clear structure and informative figures. However, some clarifications, motivations of choices, and rephrasings are advisable prior to publication. I therefore suggest the following revisions.

**Abstract**

P. 1, L. 5 and L. 12: 'proxy data': please specify 'proxy records of . . .'

P. 1, L. 7: 'with respect to data' – data in this context is very unspecific. I would suggest 'paleoproxy data'

P. 1, L. 9: I suggest replacing the vague descriptions 'different modelling choices and/or boundary conditions' for something more specific, e.g. by rephrasing to '[. . .]

different boundary conditions for climate and ice sheets, and choices for sea-ice export, formation of salty brines, and freshwater input [. . .]'

**Introduction**

P. 2, L. 25: Consider adding a reference to Galbraith and de Lavergne (2019), see also my comment for the Discussion section. Galbraith, E., de Lavergne, C. (2019). Response of a comprehensive climate model to a broad range of external forcings: relevance for deep ocean ventilation and the development of late Cenozoic ice ages. Climate Dynamics, 52(1-2), 653-679.

P. 2, L. 53: 'three sensitivity tests' - Please mention what these are e.g. 'three sensitivity tests of Southern Ocean conditions for sea-ice export, formation of brines, and freshwater input.' (see also suggestion above for the Abstract)

P. 2, L. 36-38: Model representation of Southern Ocean deep-water formation is rather central for the conclusions of this paper. I would suggest clarifying the reasons for why deep water is formed by the wrong process in most models.

**Methods**

P. 3, L. 67: There is no mention of how this sea-ice component differs from those in other PMIP models, and how the model representation of sea ice potentially impacts the results. As a non-expert on sea-ice modules, I would have liked to see a sentence or two that discusses this.

P. 3. Section 2.2: I feel that it might be clearer if this section is amended to be 'The PMIP boundary conditions and their implementation" and thus to include descriptions of the PMIP2 boundary conditions and how they differ from PMIP4 (see specific comment for L. 73-76 below for an example)

P. 3, L. 73-76: It is clearly stated (much) later in the paper (P. 10, L. 318) that GLAC-

[Figure]

1D and ICE-6G-C are the main recommendations of Kageyama et al. (2017) among multiple options. From the current phrasing here on P. 3, it is not clear why the PMIP3-option is excluded from the present study. It would be clearer if the phrasing were more similar to that on P. 10. In addition, there is no introduction to the ICE-5G option (presented on P. 4, L. 113-115), as it is part of the PMIP2 boundary conditions (see previous comment).

P. 4, Section 2.3: This section is very technical and not necessarily relevant to the average reader. I suggest moving it to an appendix, or include it with the rest of the description of the bathymetry generation method in the SI.

P. 4, Section 2.4: Please add a PI state that uses the brines parameterization, to test the effect of 'a better representation of deep-water formation' in the modern ocean.

P. 5, L. 122: Please specify why P4-I is selected as the reference LGM state over P4-G (see also comment for P. 7, L. 195)

P. 5, L. 125: 'a chosen fraction (here 0.8)' – Please specify how this choice is made and how a different choice might impact the results (see also comment for Discussion P. 11, L. 338-342)

P. 5, lines 136-139. 1) I find this paragraph to be phrased in a confusing way. I suggest separating the descriptions of LGM and PI data. 2) The MARGO Project Members reconstruct the LGM sea-surface temperatures. Please explain briefly why there is a lack of data for the Southern Ocean in austral winter. On P. 8, L. 219, you say that it is due to an extensive sea-ice cover, but coring that is done in the summer will still provide sediments from past winters, so it should be clarified why the winter sea-ice cover is a problem.

P. 5, line 137. You say here that you compare the PI simulation to World Ocean Atlas data. Please specify which version of the WOA data that is used, and if you are indeed using the WOA98, explain why you are not using the most recent version. I suspect it

is because the MARGO Project Members are using WOA98, but if so, this needs to be stated clearly. If you are using a more recent version of WOA, please cite the appropriate publications for each variable. Also, according to the figures, the PI simulation is compared to MARGO data (see comment for e.g. Fig. 4)

P. 6, L. 160: Please mention what causes this notable difference in surface area.

P. 6, L. 161-162: 'For the indicative error in the surface extent computed, we kept the respective values of 10

**Results**

P. 6, L. 178, Section 3: 'Methods'- Should be 'Results'

P. 6, L. 180: 'Cold P2 is too cold' – but it is well comparable to the more recent estimate by Tierney et al. (2020) mentioned later in the paragraph. I feel like this should be mentioned in the discussion of that paper and the fact that iLOVECLIM generally simulates more modest SAT anomalies (P. 7, L. 187-192), as this experiment design is an example of when the model actually achieves a more extreme anomaly.

P. 6-9, Section 3.2-3.3: The model-data analysis in these two sections could gain from a comparison of model skill (M) as described by Watterson (1996). This allows an evaluation of overall model-data agreement (patterns and point-to-point agreement), globally as well as on a basin level, and easier comparison between ensemble members and time periods. It would quantify statements such as those made on P. 11, L. 331-333. Watterson, I. G. (1996). Non‐dimensional measures of climate model performance. International Journal of Climatology: A Journal of the Royal Meteorological Society, 16(4), 379-391.

P. 7, L. 195: 'the reference LGM simulation P4-I' – it is never mentioned in the Methods why this simulation is chosen as the reference over P4-G. This should be clarified. Is this choice likely to influence the results of the sensitivity tests, and if so, how?

P. 7, L. 197-201: In Fig. 2, Southern Ocean SST anomalies in P4-G show a similar pattern as P4-I brines (with the exception of the mid-to-eastern Indian Ocean sector of the Southern Ocean). If you have an idea for why this is, it could be interesting to mention.

P. 8, L. 237-239: Please be clear also on how sea ice area is defined.

P. 8, L. 241: Minimum and maximum sea-ice extent - Is data available to use this method to compute corresponding numbers for the PI/modern day, to evaluate how these numbers compare to other estimates (e.g. Parkinson and Cavalieri, 2012, that is used here)? This could clarify whether the method gives rise to any systematic bias, and how well the model agrees with the different sea-ice extent estimates for the different time periods.

P.8, L. 251: The background for the previous comment is the statement here that 'the sea-ice extent of most simulations falls close to the reconstructed winter sea-ice extent'. To me, this fact seems to suggest that your maximum reconstructed extent might be an underestimation, given that most of the simulations are on the lowest end of the reference interval for glacial cooling by Annan and Hargreaves (2013).

P. 10, L. 294: 'paleotracer data' – It would be helpful to remind the reader of the relevant references.

P. 10, L. 296-298: Do you have an explanation for why the enhancement of the bottom convection cell occurs as a response to the change in ice sheet boundary conditions?

P. 10, L. 298: 'the simulation associated with GLAC-1D (compared to ICE-6G-C)' – Here, it would be helpful to specify which one of P4-G and P4-I uses which ice sheets

P. 10, L. 311-312: 'showing that simulations with a colder Southern Ocean tend to be associated with a stronger Southern Ocean cell, a weaker bottom cell and a more intense NADW cell' - Does stronger/weaker in this sense also refer to the volume occupied by the cell (i.e. depth of the water mass boundary between the bottom cell and

the NADW cell)? I get this impression when I read about the results for the 'P4-I brines' simulation. If so, it should be pointed out that proxy records conflict with this result, as they show a colder Southern Ocean simultaneously with a shallower NADW cell and a more expanded bottom cell. This is very well summarized in the Conclusions section.

**Discussion**

P. 10, Section 4.1: Important aspects of the effect of boundary conditions, modelling choices, and vertical mixing on LGM simulations are all discussed in Galbraith and de Lavergne (2018). I suggest mentioning the findings of this publication somewhere in sections 4.1-4.3.

P. 11, L. 323-324, and 326-327: Based on the remark on P. 9, L. 282, that all the simulations show similar biases in seasonal and regional patterns, could you give examples of sensitivity tests that might show somewhat different biases, or do you think this is too much of a persistent characteristic of the model (if so, why)?

P. 11, L. 338-342: It would be advisable to mention the choice of the fraction 0.8 and how it would potentially affect the results if this was chosen differently (see also comment for Methods P. 5, L. 125)

P. 11, L. 343-344: 'However, we can argue that the open ocean convection in the Southern Ocean is actually hindering the simulation of a realistic water masses distribution.' – This should be shown to be true also for the PI simulation. If it is not, the authors need to argue for why it is reasonable to include it in the LGM when it is not necessary or an improvement to do so for the PI.

P. 11, L. 349-350: 'showed that few progresses have been made by some modelling groups with respect to that aspect.' - I do not quite understand this sentence. Do you want to say "a few modelling groups have made some (minor?) progress in this aspect", or that "few modelling groups have made any progress in this aspect", or maybe that "some modelling groups have made particularly little progress in this aspect"?

P. 13, L. 409-411: 'It would therefore seem that the correct simulation of convection processes is paramount, and far more important than the choices of boundary conditions, such as the ice-sheet reconstruction [. . .]' – The brines parameterization has not been tested with the PMIP2 boundary conditions, as far as I can tell. Hence, it is clear that it is more important than the choice of ice-sheet reconstruction, but I am not sure it is well founded to say that it is more important than the choice of boundary conditions in general.

**Figures**

P. 6, L. 153: Somewhat confusing that Fig. 6 is mentioned before Fig. 2.

General comment (e.g. figures 3, 4, 7, 8, S2, S3): How are the basins defined (longitudinal and latitudinal limits)? The latitudinal limit for the Southern Ocean seems to be different in different figures (see Fig. 7).

Figure 2: Please specify in the caption what the mean SST of this simulation is, as this is specified for all other simulations in the figure. Also, it would be advisable to add a few longitude and latitude grid ticks, at least in the bottom row and left column respectively, since you have drawn the grid lines.

Figure 4, Panels a-b: In the Methods section 2.5, you say that PI simulations are compared to WOA data, not MARGO data, Caption: Describe the thinner dashed lines surrounding the 1:1 line and how the SSTs are averaged in this figure.

Figure 6, General: I would suggest adding a few longitude and latitude grid ticks, Panel a: Why is the red line in panel a dashed, when no other lines in the figure are?, Caption, L. 1: Please mention how the sea-ice edge relates to the extent and area, Caption, L. 2: The part about the arbitrary index is a bit difficult to read. I suggest "[...] as an arbitrary index on a blue to white scale, where blue denotes no indication of sea ice in proxies, and white denotes agreement of several proxies on the presence of sea ice."

Figure 7: This figure seems to have a different limit for the Southern Ocean compared to other figures (see General comment). I found this colour scale not very gentle on the eyes. I had trouble looking at the figure because the stark contrast and particularly the bright cyan/mint made me feel dizzy/nauseous. Changing it is of course not a requirement for publication, just a suggestion.

Figure 8: In my opinion, it would be preferable to plot these using a standardized grid spacing for each column, as it would make it easier to compare the slopes for the different cells

**Minor details (typos and similar)**

P. 1, L. 8: 'inaccurate' – replace for 'inaccurately'

P. 1, L. 10: 'data-model' – 'model-data' seems to be the more commonly used term, and is also what you use later in the paper

P. 1, L. 15: 'water masses properties' – replace for 'water mass properties'. Note! This error occurs throughout the manuscript when water mass properties/distributions are mentioned (see e.g. P. 2, L. 24: 'water masses distribution' – should be 'water mass distribution', as on P. 1, L. 1)

P. 2, L. 29: 'shallowing' – replace for 'shoaling'

P. 11, L. 337: 'paleodata' – replace for 'paleoproxy'

P. 13, L. 399 and 400: 'sea ice and SST data', 'proxy data' – replace 'data' by 'reconstructions'

**Supplementary information**

P. 1, first sentence: 'remplaces...' – should be 'replaces'

P. 1, fourth bullet point: 'all the critical traits stay open' – 'traits' should be 'straits'

Fig. S1: I would suggest adding a few latitude and longitude grid ticks

Fig. S2: See comments for Fig. 4

**References**

Galbraith, E., de Lavergne, C. (2019). Response of a comprehensive climate model to a broad range of external forcings: relevance for deep ocean ventilation and the development of late Cenozoic ice ages. Climate Dynamics, 52(1-2), 653-679.

Watterson, I. G. (1996). Non‐dimensional measures of climate model performance. International Journal of Climatology: A Journal of the Royal Meteorological Society, 16(4), 379-391.

---

## Author Comment (AC1) · 25 Feb 2021

We thank the reviewers for their careful reading and comments which helpfully contribute to the clarification and overall improvement of the manuscript.

We are addressing the comments below (in blue), as well as keeping track of the corrections to the text (in green). It should be noted that all mentions of line and figure numbers are made according to the numbering of the original manuscript.

**Reviewer #1:**

This paper considers a series of simulations using the iLOVECLIM model to analyze the effect of different choices of LGM boundary conditions, parameterizations, and ad-hoc modifications (to wind stress and freshwater forcing) on the LGM solution, focussing on the surface climate as well as the deep ocean overturning. Although the results are generally interesting and worthy of discussion, I think the comparison between these experiments and conclusions about which "choices" are better or more important than others need to be drawn a bit more carefully, as these various modifications aren't really comparable and some are certainly more physical than others. The paper specifically highlights the use of a parameterization to represent "sinking brines" as key to obtaining a realistic simulation of the LGM overturning circulation. However, I believe that the discussion of these results needs improvement, as elaborated below. Most importantly, I think that a PI simulations with this parameterization needs to be presented for comparison. Since the physics of the ocean have not changed between the present and LGM, the model needs to be able to reproduce the PI ocean and LGM ocean circulation with the same parameterization.

We propose here corrections on the conclusions and elements of discussion according to the reviewer's suggestions. In particular, we show the results obtained for a PI simulation with the parameterization of the sinking brines.

While we understand the reviewer's legitimate concern about the fact that our simulations are relying on modifications which are more or less physical, we would like to stress that these simulations are only sensitivity tests and do not accurately reproduce the physics of the ocean. Still, they give us the opportunity to draw conclusions about key processes, (here quoted from reviewer #2) in particular "that the sensitivity tests with winds, brines, and freshwater have more potential to influence the simulated surface properties, and particularly the ocean overturning circulation and distribution of water masses, compared to the choice of boundary conditions. This highlights the importance of informed choices in model parameterizations of processes. In particular, it further clarifies the effect of deep-water formation and convection processes for achieving a realistic representation of the glacial deep ocean and its water masses."

Our intention is certainly not to prescribe any modelling choice as key. When observing that the simulation 'P4-I brines' is the "best" one (with quotation marks, L330), we meant that it is the one that is in best agreement with geological data, given the variables analyzed in this study. We additionally qualified this simulation to be "quite crude" (L340) and we underlined it only as "one way of tackling this issue [of a simulated deep ocean circulation in disagreement with paleotracer data]" (L354). We hope that the proposed corrections will clarify this intention to the reader.

**Specific comments:**

-R1.1 : l. 15/16 in abstract. It would be good to clarify that you are referring to "different choices for LGM boundary conditions", or better yet "different choices for the LGM ice sheet topography" (see also comment below). After all, differences in boundary conditions between the PI and LGM are ultimately what has to explain the different circulation in the two climates.

Indeed. We have clarified this sentence following this comment and also the third comment of reviewer #2. This sentence now reads:

We investigate here the impact of a range of surface conditions in the Southern Ocean in the iLOVECLIM model, using nine simulations obtained with different LGM boundary conditions associated with the ice sheet reconstruction (e.g. changes of elevation, bathymetry, and land-sea mask), and/or modelling choices related to sea-ice export, formation of salty brines, and freshwater input.

Although the term "boundary conditions" might not be clear to a non-modeller reader, the change of boundary conditions due to the choice of different ice sheet reconstructions encapsulates changes of elevation, albedo, bathymetry, and land-sea mask. We prefer to use the term "ice sheet reconstruction" rather than "topography", because we are concerned that the latter might be understood essentially as a change of elevation.

Note that we have noticed a few typos along the manuscript concerning the use of a hyphen in the expression "ice sheet reconstruction", so we corrected them as well.

-R1.2 : In various places (e.g. l. 36-38) the issue of open ocean convection versus sinking along the AA slope seems to be used almost synonymously to the role of brine rejection in deep water formation, but these are rather different processes. Notably, the CCSM3 LGM simulation shows very salty AABW, clearly as a result of strong brine rejection, yet I assume AABW is still formed by open ocean convection (e.g. Shin et al. 2003, https://doi.org/10.1029/2002GL015513 - indeed this paper should be discussed).

In order to avoid confusion between the two processes, we removed the mention to brine rejection at L. 36-38.

As reviewer #2 also asked to develop this part, we included the following in the new version of the manuscript:

Moreover, Heuzé et al. (2013) showed that, even in present-day conditions, models generally simulate inaccurate bottom water temperatures, salinities and densities. Even when they do simulate relatively accurate modern bottom water properties, they still tend to form AABW via the wrong process (namely open ocean deep convection) whereas the largest proportion of AABW currently results from formation of dense shelf waters, overflowing in the deep ocean (Orsi et al., 1999; Williams et al., 2010). While some high resolution CMIP6 models now simulate dense shelf waters, Heuzé (in review, 2020) observed no obvious export of these waters, and open ocean deep convection remains a much too widespread and frequently occurring process.

We also thank the reviewer for pointing out that the Shin et al. (2003) paper is relevant for the present study. We think a mention to this reference would fit in this part of the introduction well:

And indeed, PMIP models struggle to reproduce the glacial sea-ice extent suggested by sea-ice proxy data, and especially its seasonality (Roche et al., 2012; Goosse et al., 2013; Marzocchi and Jansen, 2017). While Ferrari et al. (2014) have shown a dynamical link between the deep ocean circulation and Antarctic sea ice, Shin et al. (2003) have highlighted the major role played by Antarctic sea ice on the glacial AMOC by quantifying the haline density flux increase at the LGM in the CCSM model. Moreover, Marzocchi and Jansen (2017) have quantitatively attributed part of the observed discrepancies of the AMOC simulated by PMIP3 models to insufficient sea-ice formation and export. Therefore, targeting sea-ice biases in models may be necessary to improve the simulated water mass distribution.

-R1.3 : Relatedly, the parameterization of sinking brines needs to be described a little more– both in terms of the formulation and its physical interpretation. I understand that this method has been published previously, but since it is key to the presented conclusion I think the reader needs to be able to interpret the results from these simulations without first reading Bouttes et al. (2010). In l. 125 it is argued that "its objective is to account for the sinking of dense water along the Antarctic continental slope", and similarly in l. 301 it is referred to as "the parameterization of the sinking of dense water along the continental slope". This gives the impression that it may be a parameterization of downslopes gravity currents, which, however, seems quite misleading. Indeed,

if I understand correctly, the parameterization simply transfers salt from brine rejection directly and locally to the bottom of the ocean (without any mixing along the way), and it is not limited to the Antarctic slope. Personally, I'll have to admit that this parameterization seems rather unphysical to me (even gravity currents are associated with lots of entrainment and detrainment as they proceed down the slope, and of course they only exist on the slope). The readers can form their own opinion, but to do so, the parameterization needs to be discussed clearly.

As the expressions "along the continental slope" may indeed induce confusion with the downslope currents, we have removed them at L.125 and L.301. Also, for the reader to understand better both the formulation of this parameterization and its physical interpretation without having to refer to Bouttes et al. (2010), we expanded the following sentences as follow:

We ran 'P4-I brines' using the parameterization of the sinking of brines described by Bouttes et al. (2010). The objective of this parameterization is to account for the sinking of dense water rejected during sea-ice formation. Indeed, this process is often limited by the horizontal resolution of models, as the rejected salt tends to get diluted in the surface grid cells where sea ice is forming. This parameterization allows for a fraction of the salt content of the surface grid cell to be transferred to the deepest grid cell underneath the location of sea-ice formation. As a result, the salinity and density of the bottom cells increase while the salinity and density of the surface grid cells decrease, without congruent motion of water masses. The modification of the salinity depends on the rate of sea-ice formation, as well as the chosen fraction parameter. Here the fraction was chosen at 0.8 to allow for a large effect of this sensitivity test, but the gradual effect of this parameter choice on the streamfunction is shown in Fig. S5, as well as the impact of this parameterization on the PI streamfunction (and deep water mass properties, see Fig. S6). This simple parameterization is relatively different than a downsloping current one as it is not confined to the continental slope and does not create mixing along the way of the sinking brines. While "this brine mechanism is idealized, it reflects the impact of intense Antarctic sea-ice formation during the LGM" (Bouttes et al. (2010)) on the AABW density.

-R1.4 : As discussed in the general comments above, the "P4-I brines" simulation needs to be compared to a corresponding PI simulation with the same parameterization. Ultimately the changes in the ocean circulation between the PI and LGM climates have to be attributable to differences in the boundary conditions, not different physics. It needs to be verified that the model is able to reproduce a reasonable solution for both the LGM and PI ocean with the same parameterization. (One aspect that should be paid attention to here are the T and S properties of NADW and AABW. Importantly, AABW is fresher than NADW in the modern climate, which needs to be reproduced by any model that adequately represents water mass transformation processes in the SO.)

We agree with the reviewer. We have run a 'PI brines' simulation with the same parameter. The streamfunction obtained is presented in Fig. 1 below. We have also added it in SI for interested readers (see Fig. S5), alongside the streamfunctions of 'PI', 'P4-I' and 'P4-I brines' to enable an easier comparison, as well as other simulations that the reviewer #2 mentioned (a 'New P2 brines' and two 'P4-I brines' with a different parameter). We observe that despite the large parameter (0.8) used in 'PI brines', the differences with 'PI' remain relatively small. There is a change of the Southern Ocean overturning cell (which is expected given that the parameterization is transferring salt to the deep ocean without explicitly advecting water masses) and also a small strengthening of the bottom overturning cell in the Atlantic, but the cell limits remain similar. Therefore, we expect no significant change in the water mass distribution. As a result, we think that adding the results from 'PI brines' as a 10th simulation in all parts of the manuscript is not critical.

As for the T and S properties of deep water masses, they are now shown in Figure S6: please refer to our response to your twelfth comment for more detail.

It can also be noted that the chosen parameter for a 'PI brines' simulation (and more generally the brine parameterization) has been discussed in the reviews of Bouttes et al. (2010), which can be found at: https://cp.copernicus.org/articles/6/575/2010/cp-6-575-2010-discussion.html.

Figure 1: Streamfunctions (Sv) (as in Fig. 7) of 'PI' and 'PI brines' simulations

-R1.5 : in l. 131/132 it is argued that a quasi-equilibrium state is ensured. How is this evaluated?

We calculated the drift in time series of the global deep ocean temperature to assess when the equilibrium state is achieved – which is usually the case after 1000 years of run, though it may be a bit longer for simulations such as 'P4-I brines'. We propose clarifying this point like this: Each simulation has been run either 3000 or 5000 years to ensure a quasi-equilibrium state. The drift for any individual simulation is less that 2.10-4 °C per century for the deep ocean temperature (global mean of all oceans below 2,000 meters depth).

-R1.6 : l.161: does "surface extent" here refer to sea ice extent or still the surface of the continent? I assume the former, but please clarify.

Indeed. We have added the term "sea-ice" to avoid confusion.

-R1.7 : The error bars of 10% and 20% for winter and summer sea ice extent seem to be mostly accounting for the uncertainty in the continental margins, which is probably relatively small compared to the large uncertainty in the sea ice line. As a result, these error estimates seem very optimistic to me. Indeed this seems to be confirmed by the fact that the previous estimates of Gersonde at al. (2005) and Roche et al. (2012) fall significantly outside of this error bar. I'm not arguing that either estimate is better or worse, but simply that a larger uncertainty has to be acknowledged. I think the uncertainty range should at least encapsulate the best estimates of these major previous studies.

We here stress that both Gersonde et al. (2005) and Roche et al. (2012) used a different projection system (South Pole stereographic projection that increases distance with decreasing latitude) and subtracted the modern Antarctic ice-sheet surface to calculate the LGM winter and summer sea-ice extent instead of the LGM one. Consequently, the winter sea-ice extent of  $39 \times 10^6$  km2 and  $43.5 \times 10^6$  km2 presented in these studies was overestimated. We here used a discretization of the sea-ice line approximated on a grid with a fixed latitudinal and longitudinal spacing on a sphere and subtracted the LGM Antarctic ice sheet. We also added new control points from Benz et al. (2016)

refining the winter sea-ice extent in the Pacific sector. The recalculated area of  $32.9 \times 10^6$  km2 is therefore much more robust. Applying a similar approach to the winter sea-ice extent in Gersonde et al. (2005) and Roche et al. (2012) would provide very similar values as the one presented here. For this reason, our error bars do not need to encapsulate previous (overestimated) sea-ice extent estimates.

We however agree that error bars of 10% and 20% seem optimistic, especially as the summer seaice edge is poorly constrained from the data point of view. There is a clear lack of control points and the estimation of the sea-ice line is partly done by default (we know it is South of the marine cores with no indication of sea-ice presence, but where exactly?). We could easily double these values, though 20% seems a little pessimistic for the winter sea-ice edge which is relatively wellconstrained. To find a middle ground, we have chosen to change the error bar values to 15% and 30% in Fig. 5 and S3, and corrected the text accordingly.

However, it should remain clear for the reader that these values are only indicative. In fact, it is impossible to compute error bars in the usual sense, given the uncertainty of the reconstructed seaice edge drawn based on point data. We still wish to provide the reader with some sense of the magnitude of the error, which is why we quantified the order of magnitude of the error linked with the discretization of both this reconstructed sea-ice contour and the Antarctic continent contour at the LGM (L161-173), which adds a further element of uncertainty to the model-data comparison of the sea-ice extent. We assumed that the error associated with the sea-ice line is of the same order of magnitude as the error linked with the discretization, given the relatively coarse resolution of the CLIO model. Still, there is no statistical test which could help use quantify the error bars accurately, so the chosen values will remain questionable in any case. This is what we meant by the use of the adjective "indicative" (L161). Therefore, it would not be right to interpret how well the simulations are doing in terms of sea-ice extent based on whether they "fall inside or outside these error bars" (which is also an argument why these errors bars should not necessarily encapsulate estimates from previous studies). Note that this is an expression that we haven't used in Section 3.2, we are simply saving which simulation "falls close" to the reconstructed sea-ice extent or tend to "underestimate"/"overestimate" the sea-ice extent compared to the reconstructed one.

To stress this element further and warn the reader that these estimated error bars should not be taken at face value, we have added the adjective "indicative" in the legend of Fig. 5 as well, and modified the following sentence (L174):

Considering the order of magnitude of these alternative estimates, error bars of 15% and 30% seem reasonable. Still, these estimates are only indicative of the order of magnitude of the error.

An alternative option is, obviously, to remove all error bars. This would, however, give a false impression of certainty to the reconstructions provided.

-R1.8 : l. 220: It is argued that "the transfer of brines leads to a cooling of the Southern Ocean". Notably, however, the cooling does not occur in the regions of AABW formation but further north. There also is pronounced warming (relative to P4-I) in the North Atlantic. Do you have an explanation for these results? And does the warmer North Atlantic play a role in explaining the relatively weak and shallow AMOC in this simulation? The focus here seems to be almost entirely on the Southern Ocean, but what's the effect of the brine parameterization in the North Atlantic?

No cooling can be simulated in the regions of AABW formation due to the presence of sea ice, as the SST is at the freezing point value. The cooling is happening further north, notably in regions of upwelling and is probably due to the enhanced stratification (and decreased convection in the SO cell). We propose adding a sentence here:

We note that the transfer of salt to the bottom of the ocean leads to a cooling of the Southern Ocean ('P4-I brines', Fig. 2f), while the opposite occurs with the addition of a freshwater flux around Antarctica ('P4-I hosing', Fig. 2h). Observed in ice-free regions (i.e. where the SSTs are not necessarily at the freezing point value), this cooling is probably a consequence of the enhanced

stratification, since a well-mixed water column in upwelling regions would tend to dampen the effect of low winter surface temperatures on the SSTs.

As for the warming of the North Atlantic, it is difficult to pinpoint the exact cause of it. We only used the brine parameterization in the Southern Ocean, so it should be a consequence of the effect of this localized parameterization on the global stratification. We observe a shallower but not weaker NADW cell in 'P4-I brines', a denser AABW and fresher surface salinities. These effects should have consequences on ocean heat transport: we observe for example a slightly stronger North Atlantic drift. However, we have not looked deeper into it as this is not the bulk of our study. Indeed, we focused here essentially on the Southern Ocean. We have only used the brine parameterization in the Southern Hemisphere in 'P4-I brines', arguably because the different bathymetry affects the convection processes, but also simply to isolate the effect of a denser AABW. Though not shown in this study, we did run a simulation with the brine parameterization implemented everywhere (with the same frac=0.8) and have observed a similar structure of the AMOC, but with a weaker NADW cell and a slightly weaker bottom cell as well (compared to 'P4-I brines'), which slightly improves the model-data agreement with  $\delta^{13}$ C proxy data and the atmospheric CO2 concentration (also not shown here). We think including the results from this simulation is not really telling in the context of our study.

-R1.9 : l. 214: what is the statement that "'Cold P2' is not the simulation with the best overall agreement" based on? From what is shown in the paper, it seems to at least show among the best agreement in terms of SSTs. (And believing the Tierney et al. (2020) estimate it would also be the best in terms of global mean temperature.)

Believing the estimate from Tierney et al. (2020), 'Cold P2' is indeed best in terms of global mean surface temperature. However, considering the regional patterns of SSTs, we learn a few things. We observe in Fig. 3 (see triangles) that the simulation 'Cold P2' achieves a smaller RMSE for the Southern Ocean (and especially in the Atlantic and Indian sectors of the Southern Ocean, the region with the poorest model-data agreement) due to its lower SSTs. This is not surprising given the systematic warm bias we observe in summer around 40–50°S (Fig. 4). However, looking at the mean RMSE in winter (see crosses in Fig. 3b), we see that the RMSE for 'Cold P2' is higher than 'P4-I brines' for example: despite the lower RMSE for the Southern Ocean (Atlantic and Indian sectors, see triangles), this simulation also presents a deteriorated agreement for the Atlantic Ocean (see diamonds) in winter. This suggests that a cooling of the Southern Ocean (especially the latitudes 40–50°S of the Atlantic and Indian sectors, not so much the latitudes ~60°S of the Pacific sector) improves the model-data agreement but it is not necessarily true if this is associated with too strong a winter cooling in the North Atlantic and Arctic Ocean (see 'Cold P2' compared to 'P4-I brines' in Fig. 2). In these regions, the difference between the simulated SSTs and the SST data from MARGO Project Members (2009) can be examined in plots similar to Fig. 4 but for the Northern Hemisphere latitudes (not shown here as the focus is on the Southern Ocean).

Basically, we observe from this model-data comparison that a simulation in relatively good agreement with regional data is not necessarily the "best" simulations with respect to global estimates – and the reverse is also true. This fact is actually a limitation of the Tierney et al. (2020) data assimilation method, as their global estimate is based on the simulation with the best multi-regional agreement to data, and is not freed of potential model biases in less well-constrained regions.

Without going into too much detail, we have clarified the sentence L.214 like this:

The simulations with a colder Southern Ocean ('Cold P2', 'P4-I brines') show a better agreement with the SST data, as indicated by a smaller RMSEs computed for the Southern Ocean (see triangles in Fig. 3). However, 'Cold P2' is not the simulation with the lowest mean RMSE (see crosses in Fig. 3b), as it notably shows a higher RMSE in the Atlantic basin in winter (see diamonds).

-R1.10 : Fig. 7: What exactly is plotted here? Is it only the resolved Eulerian mean overturning or does this include the parameterized eddy transport associated with the GM parameterization? What matters for the transport of physical and geochemical tracers is really the isopycnal overturning (which probably does not have two counter-clockwise cells in SO). Computing the latter is admittedly more challenging and not commonly done in studies like this, but at least the GM contribution should be included.

We thank the reviewer for making us aware of this relevant issue. We checked how the streamfunction output of the model is computed: the contribution from the GM parameterization is indeed added to the Eulerian velocities. As for the isopycnal overturning, it has never been computed in the iLOVECLIM model and would be relatively difficult to do in a reasonable time considering our current workforce.

-R1.11 : From Fig. 7, it also seems that the AMOC in the PI simulation is too weak and too shallow, which should be discussed.

The AMOC in the PI simulation is indeed relatively weak and shallow, though it is not an outlier in the PMIP3/PMIP4 ensemble (see Fig. S1 and S2 of Kageyama et al. (in review, 2020)): the pre-industrial AMOC simulated by iLOVECLIM is fairly comparable to the pre-industrial AMOC of HadCM3, AWIESM2, MIROC-ESM and CNRM-CM5, and actually stronger and deeper than that of IPSL-CM5A2 (and IPSL-CM5A-LR).

This text has been included in the new version of the manuscript as follow:

The AMOC in our PI simulation is within the PMIP3/PMIP4 ensemble (see Fig. S1 and S2 of Kageyama et al. (in review, 2020)). In more details, it is fairly comparable to the pre-industrial AMOC of HadCM3, AWIESM2, MIROC-ESM and CNRM-CM5, and actually stronger and deeper than that of IPSL-CM5A2 (and IPSL-CM5A-LR).

-R1.12 : For the evaluation of deep ocean water mass properties it would be very useful to show T and S (as a function of latitude and depth).

Following this suggestion from the reviewer, we added plots of T and S distribution in the Atlantic ocean in Figure S6. These deep water mass properties were previously evaluated in the iLOVECLIM model under PI conditions in Bouttes et al. (2015) (Figures 6 and 7), using data from the World Ocean Atlas 2009. We reproduced the same type of plots for our 'PI' and 'PI brines' simulations, using the same data for comparison. We observe that using the brines parameterization (with the fairly high parameter choice of 0.8) warms the deep ocean interior by a few degrees, which deteriorates the agreement with WOA09 data. Moreover, while this parameterization has no significant effect on the temperature patterns at the subsurface, it causes a decrease of the surface salinity. At depth, although it lessens the salinity difference between the NADW and the fresher AABW, it also improves the salinity patterns in the North Atlantic mid-depths with respect to WOA09 data.

-R1.13 : l. 303-305: I don't follow the argument here about why 'P4-I wind' has a stronger Southern Ocean cell. My guess would be that the stronger wind stress over ice leads to enhanced ice export, which in turn leads to more new sea ice formation and thus brine rejection (c.f. Shin et al 2003).

The multiplication of the wind tension over ice indeed leads to enhanced sea-ice export and as a result increases the sea-ice area (Fig. 5), sea-ice formation and brine rejection at high latitudes. It also leads to sea ice piling up (increased sea-ice thickness). This was not clearly stated in the manuscript, which is why we propose the following modification:

On the other hand, the Southern Ocean cell is enhanced for 'P4-I wind', but moderately ('P4-I hosing') or strongly ('P4-I brines') suppressed for the other sensitivity tests. These results could be

due to the fact that the experimental setting of 'P4-I wind' – with the multiplication of the meridional wind stress on ice – enhances sea-ice export, which leads to an increased sea-ice formation and its consequent brine rejection (Shin et al., 2003).

-R1.14 : In Fig. 8 and throughout much of the manuscript "convection" seems to be used synonymously to the large-scale overturning circulation. However, convection can occur without a large-scale overturning and vice versa. I suggest to replace all references to convection cells with overturning cells.

We agree that such a terminology is more appropriate. We have changed it accordingly in Section 3.4 and in the captions of Fig. 8 and S5.

-R1.15 : In section 4.1 the various simulations are separated into those that amount to different choices for "boundary conditions" and "experimental setting", a separation that makes its way also into the abstract and conclusions. This separation, and the term "experimental setting" seems very vague. E.g. the assumed glacial temperature profile affects the simulation results via heat flux in or out of the glacier surface, and thus effectively also amounts to a difference in boundary conditions. In general it seems that "boundary conditions" is only used for cases with different choices for ice sheet topography, so I suggest to simply be explicit about that. As for the various other experiments, I don't see how they can be lumped into one category.

We also thought that the term "experimental setting" may be too vague, so thank you for the confirmation. After discussing the terminology, we find that this term is too general as it encapsulates both changes of boundary conditions (which are the changes of elevation, albedo, bathymetry and land-sea mask associated with the choice of the ice-sheet topography, such as GLAC-1D, ICE-6G-C or ICE-5G), and the modelling choices made in sensitivity tests, which concern either forcings (such as in 'Cold P2', 'P4-I hosing', 'P4-I wind'), or model parameter choices ('P4-I brines', as 'P4-I' would be with a chosen fraction of 0). This inadequate use of terminology is probably why the distinction of these different types of modelling choices seems vague. It should be clearer with the following specifications and corrections or the term "experimental setting":

- L. 72: Since the ice sheet reconstructions are still associated with large uncertainties, Kageyama et al. (2017) describe the common experimental design for LGM experiments in the current phase 4 of the project but let modelling groups choose from three different ice sheet reconstructions: GLAC-1D (Tarasov et al., 2012), ICE-6G-C (Peltier et al., 2015; Argus et al., 2014), or PMIP3 (Abe-Ouchi et al., 2015). To see the impact of such a choice, we have implemented in this study the boundary conditions (e.g. elevation, bathymetry, land-sea mask) associated with the first two options since these reconstructions are the most recent.

- L.54: "boundary conditions and other experimental settings"

- L.204: "the choice of boundary conditions and of the sensitivity tests"

- L. 295: "using different boundary conditions and/or forcing or model parameter choices (in the sensitivity tests)"

- L.286: "modelling choices"

- L. 299: "choice of forcings and model parameters"

- L.317: "relative impact of boundary conditions and other modelling choices (related to forcings or model parameter choices)"

- L.323: "the modelling choices made in sensitivity tests"

- L. 394: "sensitivity tests"

-R1.16 : Section 4.3.: Given the high uncertainty, particularly in the reconstructions of summer sea ice cover, I think it would be useful to provide some estimate of uncertainty for the sea ice seasonality from proxy data.

If we consider (as we first did) indicative error bars of 10% and 20% for the winter and summer sea-ice extent respectively, we get a seasonality estimate of  $22.7 \times 10^6$  km2 +/-  $5.3 \times 10^6$  km2. Considering higher indicative error bars of 15% and 30%, we get an estimate of  $22.7 \times 10^6$  km2 +/-  $8.0 \times 10^6$  km2, though this high uncertainty is mostly induced by the chosen error bar of 15% since the winter sea-ice extent is much larger (yet more well-constrained) than the summer one. If it was a strict estimate of the uncertainty, we would need to be more cautious with statements such as L.256-257 ("This suggests that the enhanced seasonality of the LGM Southern Ocean sea ice ( $22.7 \times 10^6$  km2 according to our proxy reconstructions, compared to the modern seasonal range of  $15.4 \times 10^6$  km2) is not entirely simulated by the model [...]"). However, as mentioned above, these estimates are only meant to give the reader a sense of the magnitude of the error, which is why we propose the following cautious addition:

First of all, the simulated seasonal amplitude of sea ice is too small with respect to the proxy data estimates, which suggest a sea-ice seasonality of  $22.7 \times 10^6$  km2 (+/-  $8.0 \times 10^6$  km2 based on 15% and 30% error bars on winter and summer sea-ice extent, respectively).

-R1.17 : I find the last paragraph of section 4.3 and specifically the attempt to reconcile the conflicting results between this study and Heuze et al. (2013) hard to follow and it seems very speculative. I don't think this discussion is necessary either, so I suggest removing this paragraph.

Having reflected on the previous text of the paragraph, we indeed found it difficult to follow in relation with the previous paragraphs. We therefore propose to modify it as follow, in the hope that it will be much clearer:

Identifying the origin of a bias is always a challenge. It might be an especially hard task to identify the origin of biases in the simulated sea-ice cover, considering the sheer number of feedbacks involved (Goosse et al., 2018). What can be noticed is that the simulated sea-ice seasonal cycle is affected by some of our modelling choices (increased in 'P4-I brines', reduced in 'P4-I wind'). Alongside, the Southern Ocean convection is suppressed in the first sensitivity test, and enhanced in the second. In a climatological mean in our model there seems to be a link between reduced Southern Ocean convection and increased sea-ice seasonal cycle. In opposition to this observation, Heuzé et al. (2013) have underlined the fact that CMIP5 models with a large sea-ice seasonality are also the ones simulating open ocean convection over extensive areas at modern times, arguing that strong sea-ice formation could precondition the ocean for open ocean deep convection. This questions the relative importance of the different simulated mechanisms at play linking the ocean convection and the sea-ice seasonal cycle, an aspect that is present in several studies (e.g. Marshall and Speer, 2012, Behrens et al., 2016; Ma et al., 2020).

-R1.18 : Based on the issues pointed out above, I'm not sure the last sentence of the conclusion can be justified. At the least, "boundary conditions, such as the ice sheet reconstruction" should be reduced to just "the ice sheet reconstruction".

We removed the term "boundary conditions" and added "for the variables analyzed in this study" to be more accurate. Still, we would like to point out that thanks to our set of simulations, we tested the impact of different types of LGM boundary conditions: ICE-5G, ICE-6G-C and GLAC-1D differ in terms of elevation (mostly), but also albedo, land-sea mask and bathymetry, whereas it is only the ocean boundary conditions which are different in the simulations 'Warm P2' and 'New P2'. As mentioned before, we are concerned that the term "ice sheet reconstruction" might be automatically associated with a change of atmosphere boundary conditions in the reader's mind, which is why we propose the following clarification:

For the variables analyzed in this study, it would therefore seem that the correct simulation of convection processes is paramount, and far more important than the choice of ice sheet reconstruction used to implement the orography and bathymetry.

We are not implying here that the convection processes simulated (or rather, parametrized) in 'P4-I brines' are "correct". This simulation is only useful to show that the intensity of the open ocean convection which is simulated in the Southern Ocean by iLOVECLIM and the large majority of models (also see Section 4.2) might be detrimental to a realistic representation of both the water mass distribution and the sea-ice seasonality. This could be of importance for other modelling groups (especially those also working on EMICs) since targeting the representation of convection processes in models may be more critical to improve the simulated glacial deep ocean circulation than spending time to implement the boundary conditions associated with different ice-sheet reconstructions in order to account for the uncertainties in the reconstructions (as suggested by Kageyama et al., 2017).

We also note that the simulation 'New P2 brines' (added on the second reviewer's suggestion in Fig. S5) yields a result similar to 'P4-I brines' (which is also the case of a simulation 'P4-G brines' not shown here), which strengthens this conclusion on the secondary impact of the choice of boundary conditions at the LGM (at least for the model resolution and variables examined in this study).

[revised manuscript text omitted]

**Supplementary information**

---

## Author Comment (AC2) · 25 Feb 2021

We thank the reviewers for their careful reading and comments which helpfully contribute to the clarification and overall improvement of the manuscript.

We are addressing the comments below (in blue), as well as keeping track of the corrections to the text (in green). It should be noted that all mentions of line and figure numbers are made according to the numbering of the original manuscript.

**Reviewer #2:**

In this paper, Lhardy et al. use an intermediate-complexity model to evaluate the response of the simulated glacial ocean circulation to Southern Ocean surface ocean temperature and sea-ice conditions. They achieve differences in these surface conditions by running simulations with various options for glacial climate boundary conditions from the Paleo Model Intercomparison Project (PMIP), and by simulations where they change wind (i.e. sea-ice export) conditions, formation of salty brines, and freshwater input in the Southern Ocean. They find that the sensitivity tests with winds, brines, and freshwater have more potential to influence the simulated surface properties, and particularly the ocean overturning circulation and distribution of water masses, compared to the choice of boundary conditions. This highlights the importance of informed choices in model parameterizations of processes. In particular, it further clarifies the effect of deep-water formation and convection processes for achieving a realistic representation of the glacial deep ocean and its water masses. This is of importance for our ability to understand, and simulate, glacial ocean storage of CO2, and to improve our models. As this category of models is commonly used for paleo simulations, this study is particularly educational for modelling groups using models of similar resolution and complexity, especially those participating in PMIP.

The experimental design is overall sound, though adding a PI state that uses the brines parameterization would be beneficial, to test the effect of 'a better representation of deep-water formation' in the modern ocean. As the parameterization does not change the amount of sea ice, only its effect on water mass properties and circulation, the choice of 0.8 as the scaling should not have to change between climate states. The amount of brines, and thus their influence on the ocean properties, should decrease in a warmer climate due to the reduction in sea-ice. It would also be useful to test the brines parameterization together with the PMIP2 boundary conditions, to strengthen some of the conclusions about the role of boundary conditions, but I leave it up to the authors to decide if this is feasible.

Overall, the paper is well written, with a generally clear structure and informative figures. However, some clarifications, motivations of choices, and rephrasings are advisable prior to publication. I therefore suggest the following revisions.

We thank the reviewer for these overall positive comments and appreciate the proofreading and meticulous suggestions provided which made our revision work easier. Following the two reviewers' comments and thanks to the relatively low computation time of iLOVECLIM, we were able to performed additional simulations with PI and PMIP2 boundary conditions with the brine parameterization (under the same model version) to answer these specific questions. We also propose corrections and elements of clarification to the points raised here.

Please note that we disagree with part of the following statement: "As the parameterization does not change the amount of sea ice, only its effect on water mass properties and circulation, the choice of 0.8 as the scaling should not have to change between climate states. The amount of brines, and thus their influence on the ocean properties, should decrease in a warmer climate due to the reduction in sea-ice". We observe that the use of the brine parameterization has an effect on sea ice, increasing both the sea-ice extent and its seasonal amplitude, and therefore further stimulating the transfer of salt to the bottom ocean. In addition, while the reduction of sea ice in a warmer climate indeed decreases the amount of brines transferred to the bottom of the ocean, a change of scaling parameter could be argued due to the impact of the higher sea level on the formation of dense water overflows on the continental shelves, as the sheer surface of these shelves is not the same for the PI. While

these reasons could potentially justify the use of a different scaling parameter (or the use of a more complex parameterization), we did keep a parameter of 0.8 in order to make the PI simulation comparable.

Finally, although we mentioned that "The parameterization of the sinking of brines is the only experimental setting we used which accommodates a better representation of both the surface conditions and the deep ocean circulation (L408-409)" (we corrected a typo here, we meant "circulation" and not "distribution"), we are not sure of where we stated that this parameterization allows for "a better representation of deep-water formation". Technically this is not true. As reviewer #1 pointed out, the use of such a parameterization does not improve the model representation of the physics of the ocean. Our objective remains simply to see the impacts of a modelling choice which accounts for a physical process which is not well represented in relatively coarse resolution models due to its subgrid scale.

**Abstract**

R2.A1 : P. 1, L. 5 and L. 12: 'proxy data': please specify 'proxy records of . . .' R2.A2 : P. 1, L. 7: 'with respect to data' – data in this context is very unspecific. I would suggest 'paleoproxy data'

We have adopted the suggested modifications.

R2.A3 : P. 1, L. 9: I suggest replacing the vague descriptions 'different modelling choices and/or boundary conditions' for something more specific, e.g. by rephrasing to '[. . .] different boundary conditions for climate and ice sheets, and choices for sea-ice export, formation of salty brines, and freshwater input [. . .]'

Following this suggestion and the first comment of reviewer #1, we have modified this sentence which now reads:

We investigate here the impact of a range of surface conditions in the Southern Ocean in the iLOVECLIM model, using nine simulations obtained with different LGM boundary conditions associated with the ice sheet reconstruction, and/or modelling choices related to sea-ice export, formation of salty brines, and freshwater input.

**Introduction**

R2.B1 : P. 2, L. 25: Consider adding a reference to Galbraith and de Lavergne (2019), see also my comment for the Discussion section. Galbraith, E., de Lavergne, C. (2019). Response of a comprehensive climate model to a broad range of external forcings: relevance for deep ocean ventilation and the development of late Cenozoic ice ages. Climate Dynamics, 52(1-2), 653-679.

We thank the reviewer for suggesting adding a reference to this article. We have done so in the Discussion section. However, we think a reference to this article at P.2 L. 25 ("Rearrangement of water masses explains part of past changes in the carbon storage capacity of the oceans (Buchanan et al., 2016; Khatiwala et al., 2019; Yu et al., 2016), which stresses the importance of correctly simulating the processes affecting the deep ocean circulation.") might not fit well with the other articles cited here. Although it is commonly accepted that the rearrangement of water masses at the LGM has had an influence on the carbon storage capacity of the ocean, the contribution of this process relative to others (changes of the biological pump efficiency, sea ice, and CO2 solubility) is still much debated. In particular, Khatiwala et al. (2019) have underlined a dominant role of the air-sea disequilibrium (and a relative small role of the ocean circulation) on the glacial CO2 drawdown. Hence the term "part of". Since Galbraith and de Lavergne (2019) did not assess the contribution of these different processes (and they could not, considering that their "biogeochemical model does

not include any air-sea effect of the variable  $pCO_2$ " and "simulates changes in dissolved nutrients and gases that are caused by the changes in ocean circulation only [...]"), we felt (but may be mistaken) that this reference would not match what is implied here by the term "part of".

R2.B2 : P. 2, L. 53: 'three sensitivity tests' - Please mention what these are e.g. 'three sensitivity tests of Southern Ocean conditions for sea-ice export, formation of brines, and freshwater input.' (see also suggestion above for the Abstract)

**We have followed this suggestion.**

R2.B3 : P. 2, L. 36-38: Model representation of Southern Ocean deep-water formation is rather central for the conclusions of this paper. I would suggest clarifying the reasons for why deep water is formed by the wrong process in most models.

Although we are unsure of the technical reasons why models are convecting too easily, we can elaborate further on Heuzé (in review, 2020) conclusions of a "too deep, too often, over too large an area" open ocean convection. Given this suggestion and the second comment of reviewer #1, we propose the following sentence:

Moreover, Heuzé et al. (2013) showed that, even in present-day conditions, models generally simulate inaccurate bottom water temperatures, salinities and densities. Even when they do simulate relatively accurate modern bottom water properties, they still tend to form AABW via the wrong process (namely open ocean deep convection) whereas the largest proportion of AABW currently results from formation of dense shelf waters, overflowing in the deep ocean (Orsi et al., 1999; Williams et al., 2010). While some high resolution CMIP6 models now simulate dense shelf waters, Heuzé (in review, 2020) observed no obvious export of these waters, and open ocean deep convection remains a much too widespread and frequently occurring process.

**Methods**

R2.C1 : P. 3, L. 67: There is no mention of how this sea-ice component differs from those in other PMIP models, and how the model representation of sea ice potentially impacts the results. As a non-expert on sea-ice modules, I would have liked to see a sentence or two that discusses this.

Goosse et al. (2013) (http://dx.doi.org/10.1016/j.quascirev.2013.03.011) have reviewed all the seaice components used in PMIP models, their table 1 is in particular informative regarding this remark. The sea-ice component in iLOVECLIM simulates a visco-plastic rheology but no ice thickness distribution, and is quite classic compared to the rest of the PMIP models. Basically, this component simulates the basic processes (thermodynamics and dynamics) but in a relatively simple way, far from the complexity of the more recently developed sea-ice component (such as LIM3.6 in Rousset et al. (2015), https://doi.org/10.5194/gmd-8-2991-2015). We propose adding the following mention:

It also includes a thermodynamic-dynamic sea-ice component described by Fichefet and Morales Maqueda (1997). This component simulates a visco-plastic rheology but no sea-ice thickness distribution, which is relatively classic compared to other PMIP models (see Table 1 of Goosse et al., 2013) but far from the complexity of more recently developed sea-ice components (Rousset et al., 2015).

R2.C2 : P. 3. Section 2.2: I feel that it might be clearer if this section is amended to be 'The PMIP boundary conditions and their implementation" and thus to include descriptions of the PMIP2 boundary conditions and how they differ from PMIP4 (see specific comment for L. 73-76 below for an example)

We have done as suggested (also see our response for the next comment).

R2.C3 : P. 3, L. 73-76: It is clearly stated (much) later in the paper (P. 10, L. 318) that GLAC-1D and ICE-6G-C are the main recommendations of Kageyama et al. (2017) among multiple options. From the current phrasing here on P. 3, it is not clear why the PMIP3 option is excluded from the present study. It would be clearer if the phrasing were more similar to that on P. 10. In addition, there is no introduction to the ICE-5G option (presented on P. 4, L. 113-115), as it is part of the PMIP2 boundary conditions (see previous comment).

Indeed. We propose the following clarification, while moving up two references (Roche et al, 2007 and Peltier, 2004):

Since the ice sheet reconstructions are still associated with large uncertainties, Kageyama et al. (2017) describe the common experimental design for LGM experiments in the current phase 4 of the project but let modelling groups choose from three different ice sheet reconstructions: GLAC-1D (Tarasov et al., 2012), ICE-6G-C (Peltier et al., 2015; Argus et al., 2014), or PMIP3 (Abe-Ouchi et al., 2015). To see the impact of such a choice, we have implemented in this study the boundary conditions (e.g. elevation, bathymetry, land-sea mask) associated with the first two options since these reconstructions are the most recent. We have also considered the results obtained with the previous LGM version of the model (PMIP2) described in Roche et al. (2007), which was generated with the boundary conditions associated with ICE-5G (Peltier, 2004), a previous reconstruction with notably higher elevation of the Northern Hemisphere ice sheets.

R2.C4 : P. 4, Section 2.3: This section is very technical and not necessarily relevant to the average reader. I suggest moving it to an appendix, or include it with the rest of the description of the bathymetry generation method in the SI.

Following your advice, we have moved this section to an appendix. To keep it near the rest of the description of the bathymetry generation method, we have also moved this part to an appendix section. As a result, only figures remain in the supplementary information. We have also changed the references to these appendices and the section numbers accordingly.

R2.C5 : P. 4, Section 2.4: Please add a PI state that uses the brines parameterization, to test the effect of 'a better representation of deep-water formation' in the modern ocean.

Please refer to our response to the fourth comment of reviewer #1. We have added an additional figure (Fig. S5) to the SI to show to the interested readers the relatively small impact of this parameterization on the PI Atlantic streamfunction.

R2.C6 : P. 5, L. 122: Please specify why P4-I is selected as the reference LGM state over P4-G (see also comment for P. 7, L. 195)

We added a sentence here:

We added to this set three sensitivity tests. The boundary conditions associated with ICE-6C-G were arbitrarily chosen as standard in these tests, which is why the simulation 'P4-I' is considered as a LGM reference in the following sections. Sensitivity tests using the simulation 'P4-G' as reference (i.e. GLAC-1D boundary conditions) yield fairly similar results (not shown here).

R2.C7 : P. 5, L. 125: 'a chosen fraction (here 0.8)' – Please specify how this choice is made and how a different choice might impact the results (see also comment for Discussion P. 11, L. 338-342)

We wanted a large effect of the brine parameterization, which is why we chose a significant fraction without it being too unrealistic (e.g. 1). The fraction values which can be considered realistic have

been largely discussed before in Bouttes et al. (2010) and in the associated peer review discussion. We have also explored different parameter values in sensitivity tests (fully comparable to the simulations in our paper: same restart and same model version), and the streamfunction obtained with frac = 0.4 and frac = 0.6 are now shown in Fig. S5. Most of the change occurs between frac = 0.4 and frac = 0.6, though a further weakening of the NADW cell and strengthening of the bottom cell can be observed between frac = 0.6 and frac = 0.8. Since the interested reader can now examine the impact of such a choice on the streamfunction in the SI, we suggest this small addition:

The modification of the salinity depends on the rate of sea-ice formation, as well as the chosen fraction parameter. Here the fraction was chosen at 0.8 to allow for a large effect of this sensitivity test, but the gradual effect of this parameter choice on the streamfunction is shown in Fig. S5, as well as the impact of this parameterization on the PI streamfunction (and deep water mass properties, see Fig. S6).

R2.C8 : P. 5, lines 136-139. 1) I find this paragraph to be phrased in a confusing way. I suggest separating the descriptions of LGM and PI data. 2) The MARGO Project Members reconstruct the LGM sea-surface temperatures. Please explain briefly why there is a lack of data for the Southern Ocean in austral winter. On P. 8, L. 219, you say that it is due to an extensive sea-ice cover, but coring that is done in the summer will still provide sediments from past winters, so it should be clarified why the winter sea-ice cover is a problem.

While moving the mention of the PI data used at the end of the paragraph in a separate sentence, we propose removing the confusing mention "due to an extensive sea-ice cover" and adding a few relevant details here:

The simulated surface conditions are first compared with the LGM sea-surface temperatures reconstructed by MARGO Project Members (2009). Thanks to the use of multiple proxies (diatoms, radiolaria, dinoflagellates, foraminifera, Mg/Ca, and alkenones), this dataset, combining 696 individual records, provides a synthesis of our knowledge of the LGM ocean surface temperature. However, it should be noted that most proxies are calibrated against summer SST (Esper et al., 2014; Cortese and Prebble, 2015) or annual SST (Sikes et al., 1997; Prahl et al., 2000). Only planktonic foraminifera allow for the estimation of winter SST (Howard and Prell, 1992) but their growth is hampered, and restricted to a couple of species, south of the Polar Front (Bé and Hutson, 1977). As such, there are only few winter SST estimates to compare with the simulated ones.

As for the model-data comparison of the PI SSTs, we relied on the modern WOA98 data (World Ocean Atlas, 1998) since it is the one used by MARGO Project Members (2009).

The additional references have also been listed in the Reference section of the paper.

R2.C9 : P. 5, line 137. You say here that you compare the PI simulation to World Ocean Atlas data. Please specify which version of the WOA data that is used, and if you are indeed using the WOA98, explain why you are not using the most recent version. I suspect it is because the MARGO Project Members are using WOA98, but if so, this needs to be stated clearly. If you are using a more recent version of WOA, please cite the appropriate publications for each variable. Also, according to the figures, the PI simulation is compared to MARGO data (see comment for e.g. Fig. 4)

We are here using WOA98 for consistency with MARGO Project Members (2009) for their modern dataset. WOA98 was chosen by MARGO Project Members (2009) because the core-top data represent average conditions (decades to centuries depending on the sedimentation rates) of sub-modern conditions (generally younger than 2000 years old) that might not be reflected by the Anthropocene climate. This is now clearly stated in the manuscript that we are using WOA98 for consistency between PI and LGM data:

As for the model-data comparison of the PI SSTs, we relied on the modern WOA98 data (World Ocean Atlas, 1998) since it is the one used by MARGO Project Members (2009).

R2.C10 : P. 6, L. 160: Please mention what causes this notable difference in surface area.

**We have completed this sentence as follows:**

To put this value into perspective, the modern Antarctic continent has a surface area of  $13.9 \times 10^6$  km2 (Fretwell et al., 2013), due to a smaller areal extension of the Antarctic ice sheet and a higher sea level.

R2.C11 : P. 6, L. 161-162: 'For the indicative error in the surface extent computed, we kept the respective values of 10

As this sentence is weirdly cut in the posted referee comment, we are not sure what the reviewer meant. But we have developed an answer to the seventh comment of reviewer #1 which concerns the chosen error bar values, maybe this provides some response to this comment as well?

**Results**

R2.D1 : P. 6, L. 178, Section 3: 'Methods'- Should be 'Results'

**Indeed! This has been corrected accordingly.**

R2.D2 : P. 6, L. 180: 'Cold P2 is too cold' – but it is well comparable to the more recent estimate by Tierney et al. (2020) mentioned later in the paragraph. I feel like this should be mentioned in the discussion of that paper and the fact that iLOVECLIM generally simulates more modest SAT anomalies (P. 7, L. 187-192), as this experiment design is an example of when the model actually achieves a more extreme anomaly.

While 'Cold P2' is too cold with respect to Annan and Hargreaves (2013), 'Cold P2' is indeed comparable to Tierney et al. (2020). However, for the reason mentioned in our response to reviewer #1 (see ninth comment), we prefer to remain cautious, especially as Tierney et al. (2020) estimate is lower than previous estimates and relies on data assimilation with the coldest model of the PMIP4 ensemble (Kageyama et al., in review, 2020). As the global mean SAT anomaly is a weighted averaged of local measurements, its reconstruction for the LGM is still debated. We have not elaborated further on the topic of Tierney et al. (2020) estimate in the discussion section than what is already mentioned L. 190-192, since the global mean SAT anomaly is not central in our study. However, a mention to this 'Cold P2' experimental design in the discussion actually strengthens our conclusion about the relative impact of boundary conditions and other modelling choices, so we thank the reviewer for pointing this out and we propose adding a sentence L.324:

In contrast, the choice of experimental setting can cause much larger differences (e.g. between 'Cold P2' and 'Warm P2', or 'P4-I' and 'P4-I brines', or 'P4-I' and 'P4-I hosing'). In particular, the differences between 'Cold P2' and 'Warm P2' suggest that, while iLOVECLIM generally simulates a more modest global SAT anomaly than other PMIP4 models (Kageyama et al., in review, 2020), modelling choices related to the glacial temperature profiles used in the radiative code can induce a very significant change.

R2.D3 : P. 6-9, Section 3.2-3.3: The model-data analysis in these two sections could gain from a comparison of model skill (M) as described by Watterson (1996). This allows an evaluation of overall model-data agreement (patterns and point-to-point agreement), globally as well as on a basin level, and easier comparison between ensemble members and time periods. It would quantify statements such as those made on P. 11, L. 331-333. Watterson, I. G. (1996). Non-dimensional measures of climate model performance. International Journal of Climatology: A Journal of the Royal Meteorological Society, 16(4), 379-391.

We are thankful for this suggestion of a metric that we did not know about. We have looked into it, in particular in comparison to other methodologies (as in Jackson et al., 2019, https://doi.org/10.1016/j.envsoft.2019.05.001). From our reading and our limited understanding of the topic, it does not seems obvious why this metric should be better than another one (cf. Jackson et al., 2019, their Table 3). Given the large amount work required to implement this methodology and the relatively small impact it could have on our results, we feel that it may not be necessary at this point. We have however contacted colleagues more directly involved in CMIP model data-model comparison to check what the accepted methods are in that community. From an initial feedback received, the M skill score does not seem to be widely used. We will continue to gather more information on this topic to better understand the potential of such a methodology (or alternative ones) for future work.

R2.D4 : P. 7, L. 195: 'the reference LGM simulation P4-I' – it is never mentioned in the Methods why this simulation is chosen as the reference over P4-G. This should be clarified. Is this choice likely to influence the results of the sensitivity tests, and if so, how?

We have chosen to clarify this point in Methods rather than in L.195, so please refer to the answer to your comment on P. 5, L. 122 for detail.

R2.D5 : P. 7, L. 197-201: In Fig. 2, Southern Ocean SST anomalies in P4-G show a similar pattern as P4-I brines (with the exception of the mid-to-eastern Indian Ocean sector of the Southern Ocean). If you have an idea for why this is, it could be interesting to mention.

It does not seem to us that SST anomalies of 'P4-G' and 'P4-I brines' have a similar pattern. For example, the warm equatorial Atlantic and cold eastern Indian in 'P4-I brines' are not evidenced in 'P4-G'. The absolute values of the anomalies are also quite different between the two simulations, who have a very different experimental design. However, we note the positive anomalies south of Africa, which can be related to the southward shift of the ACC we observed in the two simulations with respect to 'P4-I'. Although this is not the core of the paper, we added a sentence to mention this (without including figures displaying the surface currents):

The differences between 'P4-G' and 'P4-I' are small (Fig. 2e), with the exception of the eastern Atlantic and western Indian sectors of the Southern Ocean, south of the African continent, where 'P4-G' displays warmer SSTs. This positive anomaly is related to a southward shift of the Antarctic Circumpolar Current.

R2.D6 : P. 8, L. 237-239: Please be clear also on how sea ice area is defined.

**We have added the following definition:**

Only the sea-ice extent – defined as the surface with a sea-ice concentration over 15% (by convention, see the US National Snow and Ice Data Center website) – is strictly comparable to our data estimates, though we choose to present in Fig. 5 both the sea-ice extent (defined as the total surface between the northernmost 15% concentration limit and the Antarctic continent) and area (defined as the sea-ice concentration times the area of the grid cell for all ocean cells south of the equator).

R2.D7 : P. 8, L. 241: Minimum and maximum sea-ice extent - Is data available to use this method to compute corresponding numbers for the PI/modern day, to evaluate how these numbers compare to other estimates (e.g. Parkinson and Cavalieri, 2012, that is used here)? This could clarify whether the method gives rise to any systematic bias, and how well the model agrees with the different sea-ice extent estimates for the different time periods.

Winter and summer sea-ice limits were estimated through the combination of several proxies, both qualitative (relative abundances of sea-ice linked diatoms; Gersonde and Zielinski, 2000) and quantitative (Crosta et al., 2004; Esper et al., 2014). Quantitative estimates of sea ice are provided through diatom-based transfer functions that are validated on the modern model. More precisely, the modern database (composed on one hand of diatom assemblages in modern core-tops and on the other hand of modern winter sea-ice concentration or modern sea-ice duration for each modern core-tops) is run onto itself. For the modern analog technique (MAT), the transfer function selects the five most similar core-tops to each modern core-top and estimates a modern sea-ice duration for each modern core-top. It allows a direct comparison of the estimated parameter (here sea-ice duration) with the measured modern sea-ice duration (here calculated from monthly sea-ice concentration at each core-top; Crosta et al., 2004). The plot below shows that the MAT adequately reconstructs the modern distribution of sea-ice duration with no obvious regional bias as the scatter along the 1:1 line (green) is pretty constant over the whole sea-ice range (0-11 months per year). Similar results are obtained with winter sea-ice concentrations (Esper et al., 2014). Diatom-based transfer functions generally reconstruct the modern model with a root mean square error of prediction of 10% for the winter sea-ice concentration (Esper et al., 2014) and 1 month per year for sea-ice duration (Crosta et al., 2004).

Biplot of the estimated sea-ice duration (PredictedMAT) in the 257 core-tops composing the diatom modern database vs the observed sea-ice duration in the numerical atlas (SIPRES calib) at the location of the 257 core-tops composing the modern database.

R2.D8 : P.8, L. 251: The background for the previous comment is the statement here that 'the seaice extent of most simulations falls close to the reconstructed winter sea-ice extent'. To me, this fact seems to suggest that your maximum reconstructed extent might be an underestimation, given that most of the simulations are on the lowest end of the reference interval for glacial cooling by Annan and Hargreaves (2013).

While it is true that we have to be cautious with such statements given the uncertainty of the reconstructed sea-ice extents, the two following observations (the relative closeness of the simulated sea-ice extents to the winter reconstruction yet the modest global SAT anomaly of most simulations) are not necessarily inconsistent since they are on a different spatial scale. Compensation of errors may occur, for example: given the circular sea-ice distribution we observe, we think that a realistic sea-ice extent can be achieved while overestimating sea ice in the Pacific sector yet underestimating the sea-ice extent in the Atlantic sector. On a different scale, the regional biases in the Southern Ocean SSTs do not necessarily have to be consistent with a warm bias on the global scale: the modest global cooling observed in most simulations (with respect to the glacial cooling estimated by Annan and Hargreaves (2013) - and even more modest with respect to the one estimated by Tierney et al. (2020)) do not only reflect biases in the Southern high latitudes. Morevover, it should be noted that SST and sea-ice biases are not fully comparable as the tuning parameters used in the sea-ice model may affect their relationship.

R2.D9 : P. 10, L. 294: 'paleotracer data' – It would be helpful to remind the reader of the relevant references.

We have provided a few selected references here:

As Otto-Bliesner et al. (2007) have shown, iLOVECLIM is among the models which simulate a very strong glacial NADW cell at the expense of the bottom cell (as is also the case here for almost all experimental settings, see Fig. 7b,c,d,e,f,h,i), a response which is not consistent with the shallower glacial NADW and the more voluminous AABW inferred from paleotracer data (Curry and Oppo, 2005; Howe et al., 2016; Böhm et al., 2015; Lynch-Stieglitz et al., 2007).

R2.D10 : P. 10, L. 296-298: Do you have an explanation for why the enhancement of the bottom convection cell occurs as a response to the change in ice sheet boundary conditions?

Given that the surface conditions we observed for 'P4-I' and 'P4-G' are fairly similar, we think that the relatively small difference in the streamfunction may be linked to differences in land-sea mask and/or bathymetry around the coast of the Antarctic continent, which are significant in this region of large mixed layer depth (MLD). The simulation 'P4-G' tends to show deeper MLD maximums over this region compared to 'P4-I' (not shown here). As this remains hypothetical and would require including more figures, we haven't made any modification to the text.

R2.D11 : P. 10, L. 298: 'the simulation associated with GLAC-1D (compared to ICE-6G-C)' – Here, it would be helpful to specify which one of P4-G and P4-I uses which ice sheets

**The sentence now reads:**

We also notice differences between the 'P4-G' and 'P4-I' streamfunctions, with a slight enhancement of the bottom overturning cell in the 'P4-G' simulation associated with GLAC-1D (compared to the 'P4-I' simulation with ICE-6G-C) [...].

R2.D12 : P. 10, L. 311-312: 'showing that simulations with a colder Southern Ocean tend to be associated with a stronger Southern Ocean cell, a weaker bottom cell and a more intense NADW cell' - Does stronger/weaker in this sense also refer to the volume occupied by the cell (i.e. depth of the water mass boundary between the bottom cell and the NADW cell)? I get this impression when I read about the results for the 'P4-I brines' simulation. If so, it should be pointed out that proxy records conflict with this result, as they show a colder Southern Ocean simultaneously with a

shallower NADW cell and a more expanded bottom cell. This is very well summarized in the Conclusions section.

Here the adjectives stronger/weaker refer to the maximum intensity of the overturning cells and not to the volume occupied by the cell. We thought that computing the latter value would not be informative due to the lack of a bottom cell in the Atlantic basin for most simulations except for 'PI' and 'P4-I brines'. Though related, these two variables (the intensity of the overturning and the volume of the cell) are not the same, as we can observe on the 'PI' and 'P4-I brines' Atlantic streamfunctions (the volumes are relatively close, but the NADW and bottom cells are more intense in simulation 'P4-I brines'). However, in the context of this paragraph, 'P4-I brines' is actually not a good example, since the parameterization used "artificially" impacts the density of water masses (which is why we are using a subset of simulations for Fig. 8 instead of all of them, see Fig. S5). When we simulate different surface conditions in the Southern Ocean "naturally", thanks to the use of different boundary conditions, we observe this relationship between a colder SST, a stronger Southern Ocean cell, a weaker bottom cell and a stronger NADW cell. A cooling of the Southern Ocean seems associated with a deepening of the AABW/NADW boundary. And indeed, while this relationship remains true, we do have a conflict with proxy records as the simulations cannot accommodate both a better agreement with SST data in the Southern Ocean, and a better agreement with paleoproxy data which indicate a shallower NADW.

While we don't want to go into too much detail, we could add a more conclusive sentence:

The correlation coefficients R are very significant (with  $|R| \ge 0.83$  for all plots), showing that simulations with a colder Southern Ocean tend to be associated with a stronger Southern Ocean cell, a weaker bottom cell and a more intense NADW cell. While this relationship holds, modelling choices yielding colder SST in the Southern Ocean (thus in better agreement with the data) do not lead to more realistic water mass distributions. Instead, a Southern Ocean cooling seems associated with an intensification of the open ocean convection, with a negative effect on stratification.

**Discussion**

R2.E1 : P. 10, Section 4.1: Important aspects of the effect of boundary conditions, modelling choices, and vertical mixing on LGM simulations are all discussed in Galbraith and de Lavergne (2019). I suggest mentioning the findings of this publication somewhere in sections 4.1-4.3.

We thank the reviewer for the advice. Since there are a lot of findings in this article, we chose one which seems (arguably) the most relevant to provide insights (and hindsight) to our own results. We propose adding this sentence at the end of section 4.1:

It is therefore particularly important to investigate and understand the origin of these biases, while different ice sheet reconstructions have a relatively smaller impact and may not all be implemented during the PMIP4 exercise. Nonetheless, it should be noted that Galbraith and de Lavergne (2019) have investigated the effects of a broader range of forcings (greenhouse gas concentrations and orbital parameters in addition to changes in ice sheet size) on the deep water masses and they notably highlighted the nonlinear responses of their volume to varying forcings (e.g. with different global temperatures). Therefore, the choice of ice sheet reconstruction could potentially yield more significant differences in deep ocean circulation under different time periods or simulated global temperature.

R2.E2 : P. 11, L. 323-324, and 326-327: Based on the remark on P. 9, L. 282, that all the simulations show similar biases in seasonal and regional patterns, could you give examples of sensitivity tests that might show somewhat different biases, or do you think this is too much of a persistent characteristic of the model (if so, why)?

Considering the relatively coarse horizontal resolution of the iLOVECLIM model (3° x 3° for the ocean model CLIO), we do think that biases in the SST gradients (such as the one observed at 40-50°S in the Southern Ocean) are likely to remain an issue, especially as strong fronts (polar and subantarctic) are located in this region. This being said, we can still imagine sensitivity tests with a potential impact on the observed biases in seasonal and regional patterns, such as playing with the Southern Ocean westerly winds (strength, position, or effect on Ekman transport), or with the bathymetric constraint of Drake Passage: such experiments affecting winds, currents, and convection processes may impact the observed bias in surface conditions, such as the lack of interbasin contrast. In fact, we have run sensitivity tests in which we arbitrarily modified the strength and the position of the westerly winds and found a relationship between the intensity of the Southern Ocean overturning cell and the sea-ice seasonality.

R2.E3 : P. 11, L. 338-342: It would be advisable to mention the choice of the fraction 0.8 and how it would potentially affect the results if this was chosen differently (see also comment for Methods P. 5, L. 125)

We mentioned again that a model parameter choice is involved in this test by simply adding: Though legitimate, this parametrization is quite crude: a fraction (here chosen at 0.8) of the salt content of the surface grid cells is directly transferred to the deepest grid cell beneath them, without explicitly computing the convection .

As for the effects of such a choice, as shown in Fig. S5 and mentioned in our response to your comment for P. 5, L. 125, the AMOC starts to look more realistic at around frac = 0.6. However, if we analyze the model-data agreement with  $\delta^{13}$ C data (not shown in this study), the additional improvement of using frac = 0.8 is significant. Note that the reference to Figure S5 was added in the Methods rather than in the main text.

R2.E4 : P. 11, L. 343-344: 'However, we can argue that the open ocean convection in the Southern Ocean is actually hindering the simulation of a realistic water masses distribution.' – This should be shown to be true also for the PI simulation. If it is not, the authors need to argue for why it is reasonable to include it in the LGM when it is not necessary or an improvement to do so for the PI.

We agree with the reviewer that the intense open ocen convection simulated in the Southern Ocean should be detrimental for both time periods, but as the deep water formation via open ocean convection also seems to depend on the background climate (see Fig. 8), we could argue that simulating the right convection processes may seemingly be more critical to get a realistic Atlantic streamfunction at the LGM than it is at the PI. The relatively small differences in the streamfunction obtained with a "PI brines" run (see Fig. S5 and our response to the fourth comment of reviewer #1) illustrate this. After all, models are tuned at the PI, which is also why it may be possible for them to simulate reasonable bottom water properties via the wrong process – which basically amounts to an compensation of errors. As Heuzé (in review, 2020) explains: "In CMIP5 models, no model assessed by Heuzé et al. (2013) could represent dense shelf overflows correctly. Consequently, models relied on open ocean deep convection for their deep water formation. The right amount of deep convection in the Weddell Sea was required for accurate bottom properties; models that convected too little or too much were the most biased."

Still, as pointed out be reviewer #1 (and now underlined in the text, see our response to the third comment), the parameterization of brines is idealized, so we do not want to define it as default (i.e. standardly used for the PI as well). There is little open ocean convection in simulations with it since the transfer of salt to the bottom grid cells allows for dense water formation without advection being explicitly computed. This sensitivity test is useful to show that open ocean convection might be detrimental to a realistic representation of both the AMOC and the sea-ice seasonality. Nonetheless, it does not solve the issue of our model convecting too easily instead of forming dense shelf water overflows.

R2.E5 : P. 11, L. 349-350: 'showed that few progresses have been made by some modelling groups with respect to that aspect.' - I do not quite understand this sentence. Do you want to say "a few modelling groups have made some (minor?) progress in this aspect", or that "few modelling groups have made any progress in this aspect", or maybe that "some modelling groups have made particularly little progress in this aspect"?

This sentence is indeed phrased in a confusing way. We meant that Heuzé (in review, 2020) showed that some significant progress has been made by a few modelling groups, but as these modelling groups are a minority (among CMIP6 participating groups), simulating the correct convection process at the origin of deep-water formation remains an issue. We have therefore made the following modifications:

As underlined by Heuzé et al. (2013), models struggle to simulate the correct bottom water properties even in the present-day conditions, as they tend to form AABW by open ocean convection, a process rarely observed, instead of the overflow of dense continental shelf water. While none of the CMIP5 models were able to simulate the latter, Heuzé (in review, 2020) showed that a few CMIP6 models are now able to simulate AABW formation via shelf processes, notably thanks to the development of an overflow parameterization. Despite this progress, the issue remains, as "the large majority of climate models form deep water via open ocean deep convection, too deep, too often, over too large an area" (Heuzé, in review, 2020).

R2.E6 : P. 13, L. 409-411: 'It would therefore seem that the correct simulation of convection processes is paramount, and far more important than the choices of boundary conditions, such as the ice-sheet reconstruction [. . .]' – The brines parameterization has not been tested with the PMIP2 boundary conditions, as far as I can tell. Hence, it is clear that it is more important than the choice of ice-sheet reconstruction, but I am not sure it is well founded to say that it is more important than the choice of boundary conditions in general.

Please refer to our response to the last point of reviewer #1.

**Figures**

R2.F1 : P. 6, L. 153: Somewhat confusing that Fig. 6 is mentioned before Fig. 2. General comment (e.g. figures 3, 4, 7, 8, S2, S3): How are the basins defined (longitudinal and latitudinal limits)? The latitudinal limit for the Southern Ocean seems to be different in different figures (see Fig. 7).

- P6, L153: This sentence is actually not very informative at this place of the manuscript. We have removed it and instead added a reminder at the beginning of the model-data comparison of the seaice edges:

Figure 6 presents the simulated sea-ice edges alongside the sea-ice contours based on marine core data, using the reconstruction method described in Sect. 2.4. The sea-ice edge – set at 15% of sea-ice concentration by convention (NSIDC) – of all LGM simulations shows a roughly circular regional distribution around Antarctica (also see Fig. S4).

- The ocean basins are defined using a mask (see Fig. 2 below).

---

## Author Response (AR2)

We would like to thank the reviewers again for their second reading and constructive comments, as well as the editor for her helpful suggestions.
We are addressing the comments below (in blue), as well as keeping track of the corrections to the text (in green).

**Reviewer #1:**

The authors have thoroughly addressed my previous comments, and I have only two (related) minor remaining comments:

1) In l. 303-305 the authors now discuss that the strength and depth of the AMOC in the PI control simulation is broadly consistent with other PMIP simulations. However, the AMOC is too shallow compared to the real world. I still think this needs to be acknowledged.
To address this point, and following the editor's advice, we suggest completing this discussion in this manner:
The AMOC depth and strength in our PI simulation are within the PMIP3/PMIP4 ensemble (see Fig. S1 and S2 of Kageyama et al. (accepted, 2021)). In more details, the streamfunction of iLOVECLIM is fairly comparable to the pre-industrial streamfunctions of HadCM3, AWIESM2, MIROC-ESM and CNRM-CM5, and actually stronger and deeper than that of IPSL-CM5A2 (and IPSL-CM5A-LR). However, the pre-industrial AMOC strength simulated by the iLOVECLIM model is underestimated compared to modern observational data. Since 2004, the RAPID array at 26°N has measured an AMOC within the range of 13.5 Sv to 20.9 Sv, when interannual variability is accounted for (Moat et al., 2020), with a mean estimate of 17.2 Sv (McCarthy et al., 2015). The simulated AMOC strength at this latitude does not fall into this range in any of our PI simulations, which show a maximum of 10.1 Sv ('PI') and 11.2 Sv ('PI brines', Fig. S5), with both maximums occurring at depth 1225 m.
McCarthy et al. (2015) have also measured from the RAPID array a depth of the maximum AMOC generally (since they distinguished two depth modes) close to 1100 m. While we would have liked to also discuss the AMOC depth along with its strength, we find it difficult to do so considering the vertical resolution of the CLIO model at the depth of the maximum AMOC. The grid cell centered at depth 1225 m is indeed large, ranging from around 1007 to 1443 m.

2) Relatedly, in l. 367-368 it is argued that the PI4-brines simulation simulates a water mass distribution that is reconcilable with paleo proxy observations. That's an important point, but it should also be acknowledged that even with this setup the AMOC does not actually shoal between the PI and LGM simulations, which is probably inconsistent with the observational evidence.
We agree with the reviewer that stating this limitation adds a valuable nuance to this part of the discussion. Therefore, we suggest acknowledging this fact as follows:
Among our set of simulations, it is the only one simulating a water mass distribution which is reconcilable with reconstructions from paleoproxies. Nonetheless, this experimental design (like all the others tested in this study) does not result in a shoaling of the AMOC between the PI and LGM state (see Fig. S5), as is usually inferred from proxy data.

**Reviewer #2:**

In the updated manuscript, the authors have addressed the concerns of both reviewers thoroughly and in a satisfactory way, and I find their arguments for their choices to be sound. The manuscript is now clearer and easier to follow, particularly in the methods section, and the addition of a few extra simulations makes me more confident in the conclusions. I only have a few minor comments, and thus recommend publication of the article after these adjustments.

Specific comments

Line 103, parenthesis: I would put 'New P2' first in this parenthesis, to present the abbreviations in the same order as they are discussed in the text that follows. This simple change makes it easier for the reader to keep track of the different abbreviations and what they mean.

We are thankful for the reviewer's attention for details, which will surely make our paper more understandable to the reader considering the number of simulations involved. This simple change has been made.

Line 214-215: Refer here to Section 3.4, e.g. add "(see section 3.4)"

Correction implemented as suggested.

Lines 248-251: The way this is phrased now, sea-ice extent is defined twice. While the second definition is a follow-up to the first, this is not immediately clear and I had to read it multiple times to actually understand what you mean. I would recommend replacing "defined as" by "here," (or similar) in the parenthesis on line 250.

We meant for the 'second' definition to have more elements than the official – and more simply phrased – 'first' definition, in order to (1) make a clear distinction with the simulated sea-ice area defined shortly after, and (2) connect the sea-ice edge (showed in Fig. 6) to the sea-ice extent (showed in Fig. 5). We also thought that reminding the boundaries used in the computation of the sea-ice extent (sea-ice edge and Antarctic continent) might be useful for the reader to recall the methods described in Sect. 2.4 (l.160-180), which explained the source of uncertainties at play during the model-data comparison of the sea-ice extent. For these reasons, we would like to keep these elements, but we will indeed modify the phrasing as suggested to avoid confusion, as well as make a few simplifications:

Only the sea-ice extent, defined as the surface with a sea-ice concentration over 15%, is strictly comparable to our data estimates. We however chose to present both the simulated sea-ice extent (here, the total surface between the northernmost 15% concentration limit and the Antarctic continent) and area (the sea-ice concentration multiplied by the area of the grid cell for all ocean cells south of the equator) in Fig. 5.

Line 442: I still have a small issue with this sentence. The choice of the words "the correct simulation of convection processes" gives the impression that the parameterization of sinking brines indeed achieves this (even if you actually are actually rather referring to the decrease in open ocean convection). I recommend replacing "correct" by "improved".

Correction implemented as suggested.

Throughout manuscript: Check the spelling of the word 'parameterization'. It has been misspelled on several occasions.

Correction implemented as suggested.

Figure 3. Unnecessarily small font for labels, axes, and legends.

A larger fontsize is now used for Fig. 3 in the revised manuscript.

Figure 7. To illustrate some of the given (potential) explanations to the results in Section 3.4 (lines 322-324), and to connect this section more strongly to the previous about sea ice, it could be interesting to add to this figure the average sea-ice edge (given the circular shape of the sea-ice extent, the edge should be in a similar location all around the Southern Ocean, and thus the average should be reasonably representative for the entire area). This is not a requirement, simply a suggestion.

We agree that this illustration of the average sea-ice edge can be helpful to the reader, as it provides another opportunity to compare the winter and summer sea ice between simulations. When integrated to Fig. 7, these values indeed connect Sect. 3.3 and 3.4 more strongly. We have added

thin vertical lines to locate these average sea-ice edges (summer and winter) on the streamfunctions plotted in Fig. 7. The caption of the figure was modified accordingly.

[Figure]

Figure 7. Streamfunctions (Sv) in the Atlantic (North of 32°S) and Southern Ocean basins (South of 32°S). The black vertical line represents the limit between these two basins, chosen at 32°S. The thin dotted lines show the latitude of the average sea-ice edge in austral summer (red) and winter (blue) for each simulation.

Figure 8. Please specify that the grey line is a linear fit to the model results. Also, I would recommend making it slightly thicker than the grid lines.

We have modified the width of the grey lines so that they more clearly appear as black dotted lines. Before that, the thinly dotted black lines representing the linear fits did indeed look like the grey grid lines. We have also specified in the caption:

Relationships between the mean SST in the Southern Ocean (averaged up to 36°S) and the Southern Ocean (a, b), bottom (c, d) or NADW (e, f) overturning cell maximum, for all simulations except 'P4-I brines' and 'P4-I hosing'. The y-axis is inverted for the two anticlockwise cells (a, b, c, d). The dotted line represents the linear fit to the model results plotted here.

The same changes have been done to Fig. S7 in the SI.

Figure S3. Please specify that the grey line is a linear fit to the model results. Also, I would recommend making it slightly thicker than the grid lines.
Typo in second sentence 'he' should be 'The'.

The same changes as in Fig. 8 have been done to Fig. S3 and the typo is now corrected.

**Editor:**

Dear Fanny Lhardy,
I have received 2 reviews of your revised manuscript. Based on these reviews, your manuscript may be suitable for publication in Climate of the Past after some minor revisions.
Once again, both reviewers provided constructive comments and suggestions, so please address those as thoroughly as possible.
Reviewer 1 is requesting some clarifications/discussion about the AMOC for both the PI and LGM simulations and the changes (or lack thereof) between the two states; I agree that these are necessary.
When addressing their point 1, also compare the AMOC strength to available observational estimates (e.g. from the RAPID array) and explicitly report the range of the mean AMOC strength for the different experiments and how this differs from the observations.
Reviewer 2 is suggesting some additional minor changes and improvements, so please also address these.
In addition, it may also be useful to compare some of your results to the (physical) changes observed in the recently-published PI and LGM simulations of Morée et al. (CP, 2021 - doi.org/10.5194/cp-17-753-2021), where the two ocean states and model biases are evaluated against a broad range of proxy-based estimates and climate simulations.
I look forward to receiving your revised manuscript.
Best regards,
Alice Marzocchi

Dear Alice Marzocchi,

We are thankful for your work on this manuscript.
We hope we have addressed above the comments of both reviewers in a satisfactory way.

Following your complementary advice to the first comment of reviewer #1, we have used the modern AMOC strength calculated from the RAPID array (Moat et al., 2020) to quantify its underestimation in our PI simulations.

Please note that we also updated Fig. 2 (see below) on account of a computing error which was found in the mean SST calculation.

Finally, we thank you for introducing the Morée et al. (2021) study to us, which we were not aware of. The perspectives brought forward by this study are indeed insightful in the context of our paper, as the authors extensively evaluated biases, including biogeochemical ones. Although they simulate

with the NorESM-OC model an AMOC in better agreement with paleotracer reconstructions than iLOVECLIM, they still find that the remaining biases (in particular radiocarbon ages of southern sourced water, MLD at PI…) may be linked to deep water formation and convection processes in the Southern Ocean. We also find it interesting that the authors have simulated a Southern Ocean sea ice of high seasonal amplitude (their Fig. S12), and actually with a good match between the simulated winter sea-ice extent and our reconstructed estimate of $32.9 \times 10^6$ km$^2$ (as opposed to the $43.5 \times 10^6$ km$^2$ value in Roche et al., 2012). For these reasons, we would like to discuss some of the elements in Morée et al. (2021) in our Sect. 4.2, immediately following the correction related to point 2 of reviewer #1:

Among our set of simulations, it is the only one simulating a water mass distribution which is reconcilable with reconstructions from paleoproxies . Nonetheless, this experimental design (like all the others tested in this study) does not result in a shoaling of the AMOC between the PI and LGM state (see Fig. S5), as is usually inferred from proxy data. In contrast, Morée et al. (2021) were able to simulate with the NorESM-OC model a shoaled and slightly weaker AMOC at the LGM compared to their PI state. As the radiocarbon ages simulated in southern source waters were too young compared to data, they however suggested that the ventilation at the LGM was still overestimated, possibly in relation to a too small Antarctic sea-ice extent in their LGM simulation (see their Fig. S12). However, if we consider our new estimates of $10.2 \times 10^6$ km$^2$ and $32.9 \times 10^6$ km$^2$ (respectively for the summer and winter sea-ice extent inferred from proxy data), instead of the ones presented in Roche et al. (2012), the sea-ice extent simulated by Morée et al. (2021) is only slightly underestimated. Therefore additional processes might be involved to explain the weak ventilation of Southern Ocean sourced deep water at the LGM.

Sincerely,
Fanny Lhardy, on behalf of all co-authors

**References**

Moat, B. I., Smeed, D. A., Frajka-Williams, E., Desbruyères, D. G., Beaulieu, C., Johns, W. E., Rayner, D., Sanchez-Franks, A., Baringer, M. O., Denis Volkov, D., Jackson, L. C., and Bryden, H. L.: , Ocean Sci., 16, 863–874, https://doi.org/10.5194/os-16-863-2020, 2020.
McCarthy, G. D., Smeed, D. A., Johns, W. E., Frajka-Williams, E., Moat, B. I., Rayner, D., Baringer, M. O., Meinen, C. S., Collins, J., and Bryden, H. L.: Measuring the Atlantic Meridional Overturning Circulation at 26°N, Progress in Oceanography, 130, 91–111, https://doi.org/10.1016/j.pocean.2014.10.006, 2015.

---

## Author Response (AR3)

Dear Alice Marzocchi,

Thank you for your work on this paper. We are glad that it was accepted for publication.
We are addressing the minor corrections proposed below.

Thank you for submitting your revised manuscript and addressing the remaining comments from the reviewers and myself.

I am pleased to accept your manuscript for publication in Climate of the Past and only requesting you to consider a couple more minor corrections/typos that I am listing below:

Line 311: "maximums" should be "maxima"
Correction implemented as suggested.

Line 375: to be a little more accurate, rather than saying "usually", modify this to something like: "as inferred from the majority of the proxy data".
And to complete this statement, add at least a few references here of studies that suggest an AMOC weakening and shoaling of the AMOC at the LGM as inferred by proxy data (e.g. Curry and Oppo, 2005) or in addition you could also refer to some other modelling studies.
We have corrected and completed this sentence as suggested:
Nonetheless, this experimental design (like all the others tested in this study) does not result in a shoaling of the AMOC between the PI and LGM state (see Fig. S5), as inferred from the majority of the proxy data (Curry and Oppo, 2005; Böhm et al., 2015; Skinner et al., 2017; Gebbie, 2014).

In addition, please note that Anne Morée provided us with the quantitative values of LGM simulated summer and winter sea-ice extent from her recent paper. We were therefore able to implement this information into the following paragraph, instead of simply referring the their supplementary figure.
In contrast, Morée et al. (2021) were able to simulate with the NorESM-OC model a shoaled and slightly weaker AMOC at the LGM compared to their PI state. As the radiocarbon ages simulated in southern source waters were too young compared to data, they however suggested that the ventilation at the LGM was still overestimated, possibly in relation to a too small Antarctic sea-ice extent in their LGM simulation (their Fig. S12, displaying a sea-ice extent of $4.94 \times 10^6$ km$^2$ in summer and $32.95 \times 10^6$ km$^2$ in winter). However, if we consider our new estimates of $\sim 10.2 \times 10^6$ km$^2$ and $\sim 32.9 \times 10^6$ km$^2$ (respectively for the summer and winter sea-ice extent inferred from proxy data), instead of the ones presented in Roche et al. (2012), the sea-ice extent simulated by Morée et al. (2021) is underestimated only in summer.